# FormalAlign: Automated Alignment Evaluation for Autoformalization

**Jianqiao Lu**[1*]**, Yingjia Wan**[2*]**, Yinya Huang**[4]**, Jing Xiong**[1]**, Zhengying Liu**[3]**, Zhijiang Guo**[5†]
[1]The University of Hong Kong    [2]University of Cambridge
[3]Huawei Noah's Ark Lab    [4]ETH Zurich
[5]Hong Kong University of Science and Technology (Guangzhou)
jqlu@cs.hku.hk, yingjiawan.alisa@gmail.com, zhijiangguo@hkust-gz.edu.cn

## Abstract

Autoformalization aims to convert informal mathematical proofs into machine-verifiable formats, bridging the gap between natural and formal languages. However, ensuring semantic alignment between the informal and formalized statements remains challenging. Existing approaches heavily rely on manual verification, hindering scalability. To address this, we introduce FORMALALIGN, the first automated framework designed for evaluating the alignment between natural and formal languages in autoformalization. FORMALALIGN trains on both the autoformalization sequence generation task and the representational alignment between input and output, employing a dual loss that combines a pair of mutually enhancing autoformalization and alignment tasks. Evaluated across four benchmarks augmented by our proposed misalignment strategies, FORMALALIGN demonstrates superior performance. In our experiments, FORMALALIGN outperforms GPT-4, achieving an Alignment-Selection Score 11.58% higher on FormL4-Basic (99.21% vs. 88.91%) and 3.19% higher on MiniF2F-Valid (66.39% vs. 64.34%). This effective alignment evaluation significantly reduces the need for manual verification.

## 1 Introduction

Autoformalization is the task of automatically converting informal theorems and proofs into machine-verifiable formats (Wang et al., 2018; Szegedy, 2020; Wu et al., 2022; Jiang et al., 2023c). It bridges the gap between natural and formal languages, leveraging the strengths of both: natural language carries extensive logical reasoning and human knowledge. In contrast, formal language enables rigorous verification and proof, ensuring accurate and clear reasoning (Kaliszyk et al., 2014). While promising, autoformalization faces challenges in ensuring semantic alignment between these languages. The availability of fully formalized and computer-checked content is limited (Kaliszyk et al., 2017). This lack of alignment information hinders the development of robust autoformalization models (Bansal & Szegedy, 2020).

Current evaluation methods for autoformalization (Jiang et al., 2023c; Huang et al., 2024; Lu et al., 2024) focus solely on logical validity, which can be easily verified by formal language compilers (e.g., the Lean 4 compiler[1]). Another direct but suboptimal evaluation resort to surface form matching via BLEU (Papineni et al., 2002), which is widely used by recent works (Wu et al., 2022; Jiang et al., 2023c; Azerbayev et al., 2023a), but struggles with semantic alignment or logical equivalence (Li et al., 2024b).

Take the case in Figure 1 as an example, to correctly translate the natural language proof target into a Lean 4 statement, first, the variables for the objects "ligs", "lags", and "lugs" should be included and real numbers greater than zero. Then, the two equations should be translated into two corresponding hypotheses $h_1$ and $h_2$. Finally, the proof target "How many ligs are equivalent to 80 lugs? Show that 63" needs to be formalized into "$63 * a = 80 * c$", which is failed in this case by omitting the

---

[*]Leading co-authors with equal contribution.
[†]Corresponding Author.

[1]Details of the compiler are provided in Appendix A.

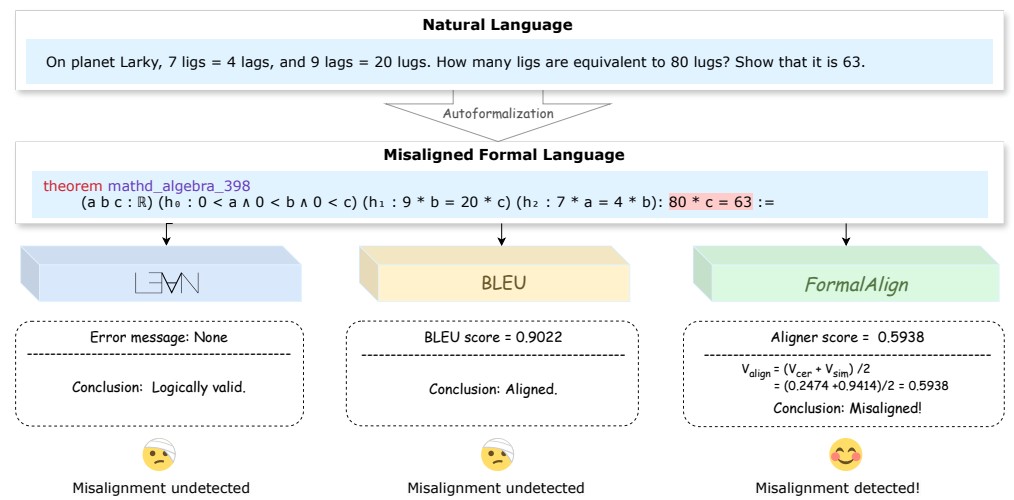

Figure 1: A comparison of current methods and FORMALALIGN in evaluating autoformalization. The formal statement is misaligned with the natural language statement: it incorrectly ends with $80 * c = 63$, when the aligned equation should be $63 * a = 80 * c$. Current methods can only verify the surface-form integrity of the autoformalized sequence via BLEU or by passing it to a formal language compiler, while our FORMALALIGN successfully detects the semantic misalignment of the autoformalized statement with the informal sequence.

pronounced "ligs". However, because the incorrectly translated "$80 * c = 63$" is logically valid in Lean 4 and similar to the ground truth in surface form, it is flawless to a theorem compiler or the BLEU score. The semantic misalignment of lacking a "lig" in the equation is undetected. Moreover, due to its elusive nature, this misalignment is often challenging to detect, even with methods like BERTscore (Zhang et al., 2020b), which are designed to assess semantic similarity, Therefore, a robust and effective approach to Automated Alignment Evaluation (AAE) is urgently needed.

To bridge this gap, we introduce the **FORMALALIGN** framework, which assesses the alignment between informal and formal languages during autoformalization. As demonstrated in Figure 2, FORMALALIGN learns both the sequence generation task of autoformalization (top half in Figure 2) and the representational alignment (bottom half in Figure 2) between input and output. FORMALALIGN jointly trains the pair of mutually enhancing tasks. This encourages the model to generate similar embeddings for corresponding pairs and distinct embeddings for non-corresponding pairs, enhancing its ability to differentiate between aligned and misaligned sequences, as the case in Figure 1.

We evaluate FORMALALIGN on four benchmarks sourced from MiniF2F (Zheng et al., 2022b) and FormL4 (Lu et al., 2024). Compared with GPT-4, FORMALALIGN achieves a substantially higher precision score across these datasets, e.g., in the FormL4-Basic dataset (93.65% vs. 26.33%). It also outperforms GPT-4 in alignment-selection score across multiple datasets, including a remarkable 99.21% vs. 88.91% in FormL4-Basic and 66.39% vs. 64.34% in MiniF2F-Valid. Extensive experimental results demonstrate the effectiveness of FORMALALIGN, significantly reducing the reliance on manual verification. Our contributions are summarized as follows:

- To the best of our knowledge, we design the **first** method for automatically evaluating alignment in autoformalization, reducing the reliance for manual verification.
- We develop a combined loss framework that simultaneously enhances a model for both autoformalization and semantic alignment.
- Extensive experiments on established autoformalization benchmarks demonstrate the effectiveness and robustness of FORMALALIGN.

## 2 RELATED WORKS

**Autoformalization** Early efforts (Wang et al., 2018; Bansal & Szegedy, 2020) employed encoder-decoder neural networks to translate informal statements into formal languages like Mizar (Rudnicki,

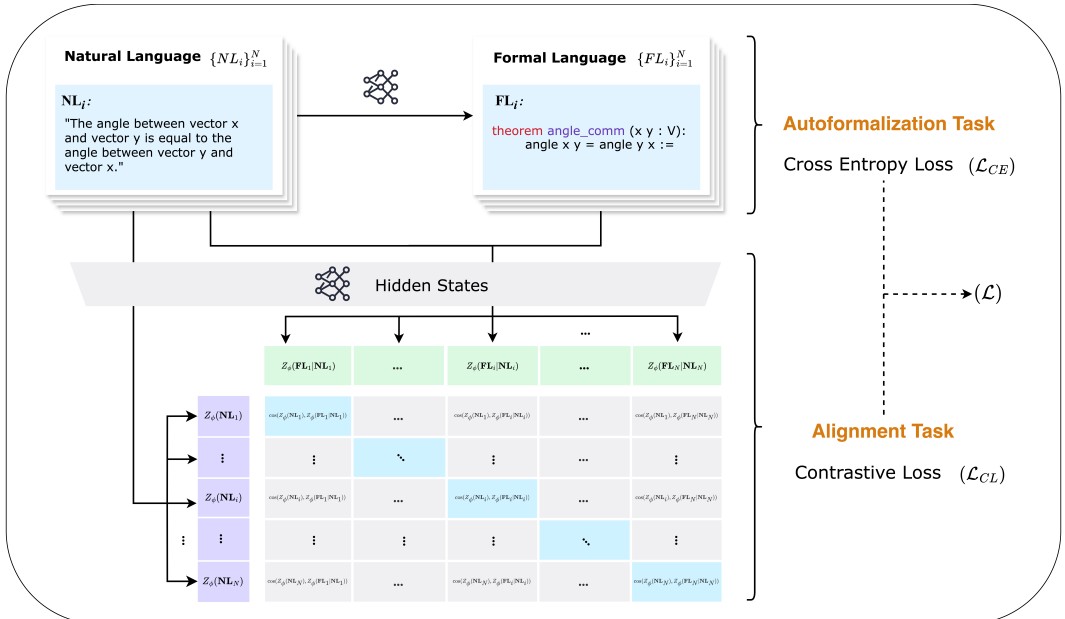

Figure 2: An overview of FORMALALIGN, which combines the cross-entropy loss in sequence autoformalization and the contrastive loss in hidden states to enhance the informal-formal alignment.

1992), HOL Light (Harrison, 1996), and Coq (Barras et al., 1997). The advent of LLMs (Chen et al., 2021; Chowdhery et al., 2022; Lewkowycz et al., 2022; Achiam et al., 2023) enhances the capabilities of autoformalization (Wu et al., 2022; Jiang et al., 2023a). Some approaches directly prompt LLMs (Wu et al., 2022; Jiang et al., 2023c; Zhao et al., 2023; Jiang et al., 2023a) to translate mathematical problems into formal languages like Isabelle (Wenzel et al., 2008) and Lean (de Moura et al., 2015). On the other hand, training or fine-tuning LLMs with paired formal-informal data (Azerbayev et al., 2023b; Ying et al., 2024; Shao et al., 2024; Lu et al., 2024) garner increasing attention for its effectiveness in enhancing LLMs' performance in autoformalization. The evaluation of autoformalization primarily depends on manually verifying the alignment between informal and formalized statements (Li et al., 2024b). There is a pressing need for an efficient and less labor-intensive method for automated autoformalization alignment.

**Text Generation Evaluation** The challenges of automatically evaluating natural language generation tasks grow as the difficulty of tasks increases. N-gram-based metrics (Papineni et al., 2002; Lin, 2004) resort to surface-form matching, which has been beneficial for evaluation tasks with specific and static references such as image-captioning (Young et al., 2014; Chen et al., 2015) and text summarization (Young et al., 2014; Cohan et al., 2018). Semantics are rarely considered until embedding-based metrics emerge, especially metrics leveraging the evolving pre-trained language models (Zhang et al., 2020a; Yuan et al., 2021; Qin et al., 2023) and LLMs (Xu et al., 2023; Jiang et al., 2023d; Liu et al., 2023b). The growth of LLMs continues empowering parameter-based metrics for advanced evaluation such as multi-agent (Chan et al., 2023) and multi-aspect (Liu et al., 2023a). The other line of work fine-tunes language models for scoring (Ke et al., 2023; Kim et al., 2023; Wang et al., 2023), labeling (Gekhman et al., 2023; Yue et al., 2023; Kim et al., 2023; Liu et al., 2023a), text probability calculation (Gekhman et al., 2023; Yue et al., 2023; Kim et al., 2023; Liu et al., 2023a), or comparison (Wang et al., 2023; Zheng et al., 2023) to enhance and adjust for evaluation targets. In this paper, we propose an automated evaluator for the challenging yet under-explored autoformalization evaluation that requires both rigorous logical validity and aligned semantics between the natural-formal pair. To this end, we fine-tune LLMs via joint autoformalization generation and representational alignment tasks and obtain a logically and semantically empowered aligner.

## 3 METHOD: FORMALALIGN

In this section, we introduce the **FORMALALIGN** framework, designed to train a FORMALALIGN model that can evaluate the alignment between natural (informal) and formal languages during autoformalization (detailed definition in Appendix M). As illustrated in Figure 2, FORMALALIGN combines two types of loss in the training process: one for the sequence generation task of autoformalization and another for the representational alignment between input and output. This dual loss framework mutually enhances autoformalization and alignment.

### 3.1 NOTATIONS

We first define the notations as follows:

- $\mathbf{NL}_i$ : The $i^{th}$ informal input sequence in a batch, $\mathbf{NL}_i = (\mathbf{NL}_{i,1}, \mathbf{NL}_{i,2}, \ldots, \mathbf{NL}_{i,m})$ , where $m$ is the sequence length of $\mathbf{NL}_i$ .
- $\mathbf{FL}_i$ : The $i^{th}$ ground-truth formal output sequence in a batch, $\mathbf{FL}_i = (\mathbf{FL}_{i,1}, \mathbf{FL}_{i,2}, \ldots, \mathbf{FL}_{i,n})$ , where $n$ is the sequence length of $\mathbf{FL}_i$ .
- $P_\phi(\mathbf{FL}_{i,j}|\mathbf{FL}_{i,<j}, \mathbf{NL}_i)$ : The probability of predicting the $j^{th}$ token in the formal sequence $\mathbf{FL}_i$ by the auto-regressive language model with parameters $\phi$ , given the previous tokens $\mathbf{FL}_{i,<j}$ in the formal sequence and the informal input $\mathbf{NL}_i$ .
- $Z_\phi(\mathbf{NL}_i)$ : The hidden state from the auto-regressive language model with parameters $\phi$ for the final position in the $i^{th}$ informal input $\mathbf{NL}_i$ , i.e., $\mathbf{NL}_{i,m}$ .
- $Z_\phi(\mathbf{FL}_i|\mathbf{NL}_i)$ : The hidden state from the auto-regressive language model with parameters $\phi$ for the final position in the $i^{th}$ ground-truth formal output $\mathbf{FL}_i$ , i.e., $\mathbf{FL}_{i,n}$ , conditioned on the paired $i^{th}$ informal input $\mathbf{NL}_i$ .
- $Z_\phi(\mathbf{FL}_{i'}|\mathbf{NL}_i)$ : The hidden state from the auto-regressive language model with parameters $\phi$ for the final position in the $(i')^{th}$ unpaired formal output $\mathbf{FL}_{i'}$ in a batch, conditioned on the $i^{th}$ informal input $\mathbf{NL}_i$ .
- $\cos(\cdot, \cdot)$ : The cosine similarity between embeddings, defined as $\cos(x, y) = \frac{x \cdot y}{\|x\| \cdot \|y\|}$ .
- $N$ : The batch size.

### 3.2 TRAINING

**Autoformalization Task** For the autoformalization task of converting an informal input sequence $\mathbf{NL}_i$ to a formal output sequence $\mathbf{FL}_i$, we use the cross-entropy loss function. This function measures the error in predicting each word in the formal sequence given the previous words and the informal input. It is defined as:

$$\mathcal{L}_{CE} = -\sum_{j=1}^{n} \log P_\phi(\mathbf{FL}_{i,j}|\mathbf{FL}_{i,j'|j'<j}, \mathbf{NL}_i)$$

**Alignment Task** To ensure that the embeddings of the informal and formal sequences are well-aligned in the FORMALALIGN model, we introduce a contrastive loss $\mathcal{L}_{\mathrm{CL}}$ . Let $\mathbf{u}_i$ and $\mathbf{v}_i$ denote the hidden state representations of the $i$ -th informal input $\mathbf{NL}_i$ and its corresponding formal output $\mathbf{FL}_i$ , respectively $\mathbf{u}_i = Z_\phi(\mathbf{NL}_i)$ and $\mathbf{v}_i = Z_\phi(\mathbf{FL}_i|\mathbf{NL}_i)$.

The contrastive loss encourages the cosine similarity $\cos(\mathbf{u}_i, \mathbf{v}_i)$ between the representations of corresponding informal-formal pairs to be higher than the cosine similarity $\cos(\mathbf{u}_i, \mathbf{v}_{i'})$ between non-corresponding pairs:

$$\mathcal{L}_{CL} = -\frac{1}{N} \sum_{i=1}^{N} \log \frac{\exp\left(\cos(\mathbf{u}_i, \mathbf{v}_i)/\tau\right)}{\sum_{j=1}^{N} \exp\left(\cos(\mathbf{u}_i, \mathbf{v}_j)/\tau\right)} \tag{1}$$

where $\tau$ is a temperature parameter that scales the cosine similarities. By minimizing this contrastive loss, the FORMALALIGN model learns to align the embeddings of corresponding informal-formal sequences while ensuring that the embeddings of non-corresponding sequences are dissimilar.

**FORMALALIGN Loss**   We jointly train an evaluator model with the autoformalization task and the alignment task, resulting in a FORMALALIGN model. The combined training loss is:

$$\mathcal{L} = \mathcal{L}_{CE} + \mathcal{L}_{CL} \tag{2}$$

We train an alignment-aware FORMALALIGN model by minimizing a combined loss, enabling it to benefit from both the sequence alignment inherent in the autoformalization and the representation alignment facilitated by the contrastive learning process.

## 3.3   INFERENCE

During the inference phase, the FORMALALIGN model generates an alignment evaluation score $\mathcal{V}_{\text{align}}$ for each pair of informal input $\mathbf{NL}_i$ and formal output $\mathbf{FL}_i$ . This score is a combination of two metrics: the certainty score and the similarity score.

**Certainty Score**   The certainty score $\mathcal{V}_{\text{cer}}$ measures the confidence of the fine-tuned FORMALALIGN model in predicting the formal output based on the corresponding informal input. It is calculated by taking the exponential of the average log-probability assigned by the model to each token in the formal sequence:

$$\mathcal{V}_{\text{cer}} = \exp\left(\frac{1}{n}\sum_{j=1}^{n}\log P_\phi(\mathbf{FL}_{i,j}|\mathbf{FL}_{i,<j}, \mathbf{NL}_i)\right) \tag{3}$$

where $P_\phi$ represents the probability output of the model with parameters $\phi$ , $\mathbf{FL}_{i,<j}$ denotes the tokens in the formal sequence up to position $j-1$ , and $n$ is the length of the formal sequence.

**Similarity Score**   The similarity score $\mathcal{V}_{\text{sim}}$ measures alignment between the embedding representations of the informal input and the formal output. It is computed using the cosine similarity between the hidden states of the informal input and the formal output conditioned on the informal input:

$$\mathcal{V}_{\text{sim}} = \cos(Z_\phi(\mathbf{NL}_i), Z_\phi(\mathbf{FL}_i|\mathbf{NL}_i)) \tag{4}$$

where $Z_\phi(\mathbf{NL}_i)$ represents the hidden state from the final position in the informal input, and $Z_\phi(\mathbf{FL}_i|\mathbf{NL}_i)$ represents the hidden state from the formal output conditioned on informal input.

**Alignment Score**   The overall alignment evaluation score $\mathcal{V}_{\text{align}}$ is computed by taking the average of the certainty score and the similarity score:

$$\mathcal{V}_{\text{align}} = (\mathcal{V}_{\text{cer}} + \mathcal{V}_{\text{sim}})/2 \tag{5}$$

This combined score reflects both the accuracy of the translation from informal to formal expressions and the alignment of the internal representations of the sequences, providing a robust evaluation metric during the inference stage.

## 4   EXPERIMENT

### 4.1   DATASETS

In our experimental setup, we conduct fine-tuning on the FormL4 (Lu et al., 2024) and MMA (Jiang et al., 2023a) training sets, both of which are derived from Mathlib, a library of fundamental mathematical statements. This training data enables our model to align informal mathematical statements with their formal counterparts.

To thoroughly evaluate our method's ability to align informal mathematical statements with formal language, we employ a comprehensive set of test sets that covers both in-domain and out-of-domain data. Specifically, we use four distinct test sets: the basic and random test sets from FormL4, and the valid and test sets from MiniF2F (Zheng et al., 2022a). FormL4, designed to assess the autoformalization capabilities of LLMs in Lean 4 (de Moura & Ullrich, 2021) sourced from Mathlib, provides a comprehensive evaluation framework. The basic and random test sets from FormL4 allow

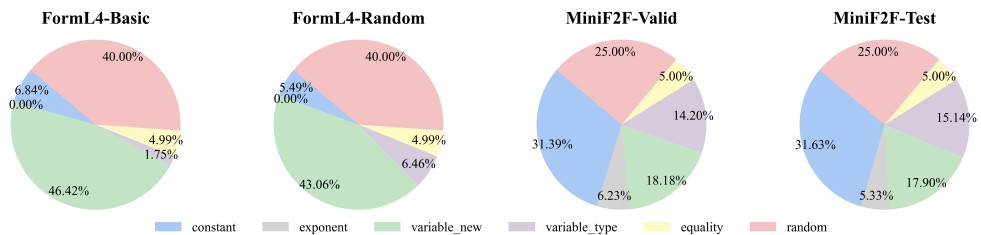

Figure 3: Distribution of Misalignment Types across Four Datasets. This figure illustrates the variety and proportion of misalignment strategies applied to generate negative examples in the FormL4-Basic, FormL4-Random, MiniF2F-Valid, and MiniF2F-Test datasets, providing a comprehensive evaluation basis for the AAE task.

us to gauge the model's performance in autoformalizing fundamental math statements that are similar to the training data. In contrast, the validation and test sets from MiniF2F serve as out-of-domain test data, providing a more challenging evaluation setting. MiniF2F is a benchmark containing 488 manually formalized mathematical competition statements sourced from various mathematical olympiads (AMC, AIME, IMO) and high-school and undergraduate math classes.

These datasets primarily provide paired input-output instances, lacking the negative examples crucial for a more robust assessment of our model. Consider one aligned informal-formal pair shown in Table 1 as an example. We detail our approach to generating misaligned formal outputs with the natural (informal) input employing strategies outlined in Table 2. The distribution of these misalignment types is visualized in Figure 3.

Table 1: Natural Language Statement and its aligned Lean Formal Statement.

**Natural Language Statement**

The volume of a cone is given by the formula $V = \frac{1}{3}Bh$, where $B$ is the area of the base and $h$ is the height. The area of the base of a cone is 30 square units, and its height is 6.5 units. What is the number of cubic units in its volume? Show that it is 65.

**Lean Formal Statement**

```
theorem mathd_algebra_478
  (b h v : ℝ)
  (h₀ : 0 < b ∧ 0 < h ∧ 0 < v)
  (h₁ : v = 1 / 3 * (b * h))
  (h₂ : b = 30)
  (h₃ : h = 13 / 2) :
  v = 65 :=
```

## 4.2 METRICS

To assess the performance of models in evaluating the alignment of informal and formal language pairs, we introduce three automated metrics:

**Alignment Selection (AS)** This metric quantifies how well a model selects the aligned formal output from multiple candidates when given an informal input. We calculate the alignment evaluation score $\mathcal{V}_{\text{align}}$ with details described in Appendix C.3 for each informal-formal pair. The pair with the highest score is selected as the aligned pair.

**Alignment Detection** We introduce a predefined threshold $\theta$ to detect the alignment for each informal-formal pair. If $\mathcal{V}_{\text{align}}$ exceeds $\theta$, the model considers the pair to be aligned. We evaluate this

Table 2: Misalignment Strategies.

| Misalignment Strategies | |
|---|---|

**Constant Modification (constant)**
This type of misalignment involves changing a constant value within the expression.

```
theorem mathd_algebra_478
 (b h v : ℝ)
 (h_0 : 0 < b ∧0 < h ∧0 < v)
 (h_1 : v = 1 / 3 * (b * h))
 (h_2 : b = 31) -- changed constant
 (h_3 : h = 13 / 2) :
 v = 65 :=
```

**Exponent Modification (exponent)**
This misalignment targets the exponents in the expression.

```
theorem mathd_algebra_478
 (b h v : ℝ)
 (h_0 : 0 < b ∧0 < h ∧0 < v)
 (h_1 : v = 1 / 3 * (b^2 * h)) --
     changed exponent
 (h_2 : b = 30)
 (h_3 : h = 13 / 2) :
 v = 65 :=
```

**Introduction of a New Variable (variable_new)**
This misalignment introduces a completely new variable into the expression.

```
theorem mathd_algebra_478
 (b h v x : ℝ) -- added a new
     variable x
 (h_0 : 0 < b ∧0 < h ∧0 < v)
 (h_1 : v = 1 / 3 * (b * h))
 (h_2 : b = 30)
 (h_3 : h = 13 / 2) :
 v = 65 :=
```

**Change of Variable Type (variable_type)**

In this case, the misalignment involves changing the type of a variable within the expression. The function identifies the type of a randomly selected variable and changes it to a different type from a predefined list of types.

```
theorem mathd_algebra_478
 (b h v : ℤ) -- changed type to ℤ
 (h_0 : 0 < b ∧0 < h ∧0 < v)
 (h_1 : v = 1/3 * (b * h))
 (h_2 : b = 30)
 (h_3 : h = 13/2) :
 v = 65 :=
```

**Modification of Equality (equality)**
This misalignment switches between equality = and inequality ≠ symbols within the expression.

```
theorem mathd_algebra_478
 (b h v : ℝ)
 (h_0 : 0 < b ∧0 < h ∧0 < v)
 (h_1 : v ≠1 / 3 * (b * h)) --
     swapped inequality
 (h_2 : b = 30)
 (h_3 : h = 13 / 2) :
 v = 65 :=
```

**Random Pairing (random)**
This creates a mismatch between the informal input and its formal output. Instead of pairing the informal input with its correct formal output, this strategy randomly selects a formal output from other examples.

detection method using two metrics: precision and recall. Firstly, the **Precision** metric measures the fraction of pairs identified as aligned by the model that are truly informal-formal pairs. It is calculated as Precision $= \frac{TP}{TP+FP}$, where $TP$ represents the number of true positives (correctly identified aligned pairs) and $FP$ represents the number of false positives (incorrectly identified aligned pairs). Secondly, the **Recall** metric measures the fraction of true informal-formal pairs correctly identified by the model. It is calculated as: Recall $= \frac{TP}{TP+FN}$, where $FN$ represents the number of false negatives (missed aligned pairs).

## 4.3 MAIN RESULTS

We fine-tune a Mistral-7B model (Jiang et al., 2023b) as the FORMALALIGN model and evaluate its performance on various autoformalization benchmarks. The datasets used in this study include FormL4-Basic, FormL4-Random, MiniF2F-Valid, and MiniF2F-Test. Each data example consists

Table 3: Automated Alignment Evaluation (AAE) results across different autoformalization benchmarks. The table compares the performance of our fine-tuned Mistral-7B model (FORMALALIGN model) with GPT-4 and GPT-3.5 on four datasets: FormL4-Basic, FormL4-Random, MiniF2F-Valid, and MiniF2F-Test. Performance metrics include Alignment Score (AS), Precision (Prec.), and Recall (Rec.).

| Datasets | FormL4-Basic | | | FormL4-Random | | | MiniF2F-Valid | | | MiniF2F-Test | | |
|---|---|---|---|---|---|---|---|---|---|---|---|---|
| | AS | Prec. | Rec. | AS | Prec. | Rec. | AS | Prec. | Rec. | AS | Prec. | Rec. |
| GPT-4 | 90.23 | 42.68 | 88.15 | **91.85** | 45.72 | **89.95** | 67.24 | 59.85 | **89.87** | **70.82** | 62.45 | **92.88** |
| GPT-3.5 | 50.23 | 25.21 | **90.83** | 47.00 | 23.42 | 67.26 | 47.32 | 22.29 | 62.55 | 40.74 | 21.97 | 61.73 |
| FORMALALIGN | **99.21** | **93.65** | 86.43 | 85.85 | **86.90** | 89.20 | **66.39** | **68.58** | 60.66 | 64.61 | **66.70** | 63.37 |

of an aligned informal-formal pair, which is considered a positive example. To comprehensively assess the model's performance and robustness, we augment each positive example with 21 negative examples generated through carefully designed misalignment strategies outlined in Table 2.

To balance precision and recall in the FORMALALIGN model's alignment detection, we set $\theta = 0.7$. Table 3 presents the detailed experimental results, including Alignment-Selection (AS), Precision (Prec.), and Recall (Rec.) metrics. The table compares the performance of our fine-tuned Mistral-7B model (FORMALALIGN model) with GPT-4 (Achiam et al., 2023) and GPT-3.5 (OpenAI, 2023) across the different datasets. For more information on the query prompts used in the experiments, please refer to Appendix C.2.

**Effective and Robust Alignment Evaluation:** The experimental results demonstrate the effectiveness and robustness of our FORMALALIGN in evaluating the alignment between informal and formal languages. The model achieves impressive performance, with high alignment, precision, and recall scores across all datasets. Notably, on the FormL4-Basic dataset, it attains an exceptional Alignment-Selection score of 99.21% and a Precision of 93.65%. These results highlight the model's ability to accurately identify aligned informal-formal pairs.

**Generalization Across Datasets:** The FORMALALIGN model exhibits consistent performance across four diverse datasets, demonstrating its ability to generalize its autoformalization evaluation capabilities. Particularly noteworthy are the model's AS scores of 66.39% and 64.61% on the challenging MiniF2F-Valid and MiniF2F-Test datasets, respectively. These scores are comparable to those achieved by GPT-4, which obtained AS scores of 64.34% and 68.31% on the same datasets. The FORMALALIGN model's strong performance on the MiniF2F theorem proving benchmark, which poses significant challenges due to its complexity and diversity, highlights the effectiveness of our proposed FORMALALIGN in enhancing the model's generalization ability.

The experimental results validate the effectiveness of FORMALALIGN in improving the performance of LLMs for autoformalization alignment evaluation. The integration of cross-entropy loss with contrastive learning in the model's training process has proven to be a powerful combination, resulting in a robust model capable of achieving high alignment-selection, precision, and recall scores across various datasets. The model's ability to generalize its performance to challenging benchmarks like MiniF2F further underscores the benefits of our approach.

## 4.4 COMPARISON WITH HUMAN EVALUATION AND LLM-AS-JUDGE

To comprehensively assess our FORMALALIGN model in autoformalization alignment evaluation, we conduct an extensive human evaluation along with an LLM-as-judge evaluation and compared their correctness rates. This analysis offers an in-depth understanding of our automated evaluation method's performance compared to human experts and state-of-the-art language models.

The experiment design, statistical results, as well as detailed discussions are specified in Appendix G. As listed in G, human experts achieved the highest correctness ratio in matching with the ground-truth alignment evaluations with an average of 79.58%, followed by our FORMALALIGN (65.00%). The LLM-as-judge method achieves the lowest precision in autoformalization alignment evaluation. Each human expert takes approximately 3 hours to review 80 items, while the FORMALALIGN model requires less than 2 minutes to conduct the automated evaluation. These findings emphasize the value

of our FORMALALIGN framework in providing an efficient and reliable automated evaluation method for autoformalization alignment.

# 5 ANALYSIS AND DISCUSSION

To further validate the robustness and effectiveness of our FORMALALIGN framework, we conducted seven additional experiments, some of which are detailed in the Appendix due to limited space. We begin by validating the generalized effect of our FORMALALIGN across different baseline language models (Section 5.1). We investigate the necessity and effectiveness of our combined training loss (Section 5.2) and the impact of our proposed alignment score $\mathcal{V}_{\text{align}}$ (Section 5.3).

Furthermore, we address concerns regarding potential data contamination in pre-trained language models through a comprehensive analysis of our experimental data (Appendix D). Next, we investigate the generalization ability of our method and the impact of different training datasets on its performance (Appendix E). We then explore the effect of incorporating contrastive learning loss on the performance of autoformalization of natural language statements to formal language statements (Appendix F). To ensure the comprehensiveness and true representation of potential misalignments, we conduct an extensive manual review and evaluation of our FORMALALIGN framework (Appendix G).

## 5.1 EFFECTS OF DIFFERENT BASELINES

In this section, we validate the generalized effect of our FORMALALIGN across different baseline language models. These baselines are Phi2-2.7B (Javaheripi et al., 2023) (**Phi**), LLaMA2-7B (Touvron et al., 2023) (**LLaMA**), DeepSeekMath-Base 7B (Shao et al., 2024) (**DeepSeek**) and Mistral-7B (Jiang et al., 2023b) (**Mistral**). Table 4 presents the Alignment-Selection performance of the different baseline models across four datasets:

Table 4: Alignment Selection Performance of different baselines across 4 datasets.

| Datasets | FormL4 | | MiniF2F | |
|---|---|---|---|---|
| | **Basic** | **Random** | **Valid** | **Test** |
| **Phi** | 80.77 | 71.07 | 31.56 | 32.51 |
| **DeepSeek** | 90.29 | 77.08 | 54.66 | 55.19 |
| **LLaMA** | 98.08 | 76.42 | 54.51 | 57.20 |
| **Mistral** | 99.21 | 85.85 | 66.39 | 66.70 |

The experimental results indicate that the Mistral model outperforms the other baseline models across all datasets, demonstrating the highest Alignment-Selection performance. The LLaMA and DeepSeek models perform strongly, particularly on the FormL4 datasets. We note that the Phi model still performs adequately on the FormL4 datasets but struggles on the MiniF2F datasets, highlighting that our method is easily applicable to smaller models, as Phi2 has less than half the parameters compared to the other three models.

These results validate the effectiveness of our FORMALALIGN in improving the automated alignment evaluation performance across various baseline language models, with Mistral showing the most significant improvements. This suggests that our FORMALALIGN can generalize well across different model architectures.

## 5.2 EFFECTS OF DIFFERENT TRAINING LOSS

We investigate the necessity and effectiveness of our combined training loss, defined in Eq. 2, by conducting an ablation study with different loss configurations. The results, presented in Table 5, provide valuable insights into the impact of each loss component on the model's performance.

The configuration using only the cross-entropy loss (**w/ CE**) achieves comparable performance, particularly on the FormL4 dataset. This result suggests that the autoformalization task, optimized by the cross-entropy loss, inherently learns alignment between informal and formal sequences.

Table 5: Comparison of overall alignment-selection performance across different configurations: with only cross-entropy loss (w/ CE), with only contrastive loss (w/ CL), and the complete model (Ours).

| Datasets | FormL4 | | MiniF2F | |
|---|---|---|---|---|
| | Basic | Random | Valid | Test |
| w/ CE | 98.64 | 82.81 | 52.45 | 54.32 |
| w/ CL | 59.05 | 57.55 | 36.07 | 30.86 |
| Ours | **99.21** | **85.85** | **66.39** | **66.70** |

Although the configuration using only the contrastive loss (**w/ CL**) shows limited performance, it plays a crucial complementary role to the cross-entropy loss. The combined approach (**Ours**), which incorporates both cross-entropy and contrastive losses, achieves the best performance across all datasets.

## 5.3 EFFECTS OF DIFFERENT ALIGNMENT SCORE $\mathcal{V}_{\text{ALIGN}}$

We investigate the necessity of our proposed alignment score $\mathcal{V}_{\text{align}}$ as described in Eq. 5. Table 6 provides a comprehensive evaluation of the effectiveness of our proposed alignment score $\mathcal{V}_{\text{align}}$. By analyzing different configurations of the model, we derive the following key insights:

Table 6: Comparison of overall alignment-selection performance across: with only the certainty score (w/ cer), with only the similarity score (w/ sim), and the complete model (Ours).

| Datasets | FormL4 | | MiniF2F | |
|---|---|---|---|---|
| | Basic | Random | Valid | Test |
| w/ cer | 98.98 | 85.64 | 53.69 | 55.55 |
| w/ sim | 45.25 | 20.75 | 20.49 | 21.81 |
| Ours | **99.21** | **85.85** | **66.39** | **66.70** |

The configuration using only the certainty score (**w/ cer**) achieves high performance, particularly on the FormL4 dataset. This result indicates that the model's language generation capabilities are robust and significantly contribute to the alignment evaluation. The certainty score measures the model's confidence in predicting the formal output, underscoring the importance of accurate language generation in our method.

While using only the similarity score (**w/ sim**) shows limited performance, the combined approach (**Ours**), which integrates both certainty and similarity scores, achieves the best result across all datasets. This result demonstrates that combining both scores provides a more holistic and reliable evaluation metric. The combined score captures language-based and representation-level information, ensuring a robust evaluation during inference.

## 6 CONCLUSION

In this study, we introduce FORMALALIGN, a framework designed to automate the alignment evaluation in the autoformalization process using LLMs. Our approach utilizes a dual loss function that combines cross-entropy and contrastive learning loss, significantly enhancing the model's ability to discern and align informal-formal language pairs. This methodology not only preserves the integrity of logical constructs but also improves the accuracy of alignment between informal and formal sequences. Extensive experiments conducted across four datasets demonstrate that FORMALALIGN effectively reduces the reliance on manual verification processes, thereby streamlining the autoformalization workflow. The results confirm that our method provides reliable, effective, and robust evaluations, proving its practical utility in real-world scenarios. We believe that FORMALALIGN opens new avenues for research and application in the autoformalization field, offering a scalable and efficient solution to one of the most pressing challenges in the domain.

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

## A    LEAN 4 COMPILER

The Lean 4 Compiler is a critical component of the Lean 4 programming language. This tool enables users to craft effective proof automation tactics within the Lean environment and transform them into optimized C code. The Lean 4 Compiler in our scope is referred to as the tool available at `https://github.com/leanprover-community/repl`. This particular resource provides a read-eval-print loop (REPL) designed for Lean 4, which supports user interaction through JSON formatted input and output streams (stdin and stdout, respectively). Our compilation projection is therefore founded on REPL.

## B    MORE RELATED WORKS

**Formal Mathematics**    Formal languages, such as Isabelle (Wenzel et al., 2008), Lean (de Moura & Ullrich, 2021), HOL Light (Harrison, 1996), and Coq (Barras et al., 1997), have become integral tools in modern mathematics verification systems. These interactive theorem provers (ITPs) function as programming languages, allowing users to input statements and proofs in a formal language for automatic correctness verification. Recent work has explored various approaches to automate the formalization process across different contexts. Murphy et al. (2024) developed a neuro-symbolic framework for Euclidean geometry that specifically addresses diagram-dependent proofs, using theorem provers to incorporate diagrammatic information into the formalization process. Taking a different approach, Li et al. (2024a) focused on improving candidate selection in LLM-generated formalizations through symbolic equivalence and semantic consistency verification. At a larger scale, Zhang et al. (2024) tackled library-level consistency challenges through retrieval-augmented generation, denoising, and syntax error feedback. While these works address different aspects of formalization - from geometric reasoning to candidate selection to library-scale consistency - they highlight the diverse challenges in bridging informal and formal mathematics.

## C    EXPERIMENTAL DETAILS

### C.1    FINETUNING DETAILS

Our experiments are conducted in a computing environment with 8 NVIDIA A100 GPUs, each with 40GB of memory. All models are fine-tuned in a full-parameter setting.

We employ the AdamW optimizer for model training over 1 epoch, with a batch size of 512. The learning rate is set at $2e \times 10^{-6}$, incorporating a 3% learning rate warmup period. Below, we present a comprehensive overview of the training hyperparameters utilized. These parameters are consistently applied across training all LLMs.

### C.2    PROMPT DETAILS

In this section, we provide two key prompts used to evaluate the alignment between informal mathematical inputs and their formalized theorem outputs. The first prompt is designed for querying

Table 7: Finetuning Hyperparameters.

| Hyperparameter | Value |
|---|---|
| Global Batch Size | 128 |
| LR | $5 \times 10^{-6}$ |
| Epo. | 1 |
| Max Length | 2048 |
| Weight Decay | 0 |
| Warmup Ratio | 0.03 |

GPT-based models using explicit scoring mechanisms, while the second prompt demonstrates how our FORMALALIGN model computes alignment scores through a computational framework enabled by its combined loss training.

We report greedy decoding results for GPT-4 and GPT-3.5 using a temperature setting of 0.0. Additionally, For the GPT-3.5 version, we query the API of gpt-3.5-turbo-0125. For GPT-4, we query the API of gpt-4-1106-preview.

**Prompt for Querying GPT for Automated Alignment Evaluation (CoT-based)**

This prompt guides GPT-4 through a step-by-step reasoning process to identify potential discrepancies and evaluate alignment systematically.

Below, we provide the CoT-based prompt used to query GPT for automated alignment evaluation:

```
Given an informal mathematical input and a formal theorem statement, your task is to
    evaluate the alignment between them. Assign a binary value (0 or 1) to each formal
    output, where:

— 0 indicates that the formal output does not align with the informal input.
— 1 indicates that the formal output aligns with the informal input.

To ensure a thorough and accurate evaluation, follow these steps:
1. Understand the Informal Input: Identify the key semantic and structural elements in
    the informal input.
2. Analyze the Formal Output: Compare the formal output against the informal input
    based on the following criteria:
   — Semantic Consistency: Does the formal output accurately capture the meaning of the
       informal input?
   — Structural Correspondence: Does the structure of the formal output reflect the
       structure implied in the informal input?
   — Completeness: Does the formal output include all relevant information from the
       informal input?
   — Precision: Is the formal output free from extraneous or incorrect information that
       is not present in the informal input?
3. Identify Discrepancies: Note any mismatches or issues based on the criteria above.
    Use specific reasoning to justify your assessment.
4. Decide on Alignment: Based on your step—by—step analysis, assign a value of 0 (not
    aligned) or 1 (aligned) to each formal output.

Your response should include:
1. A brief explanation of your reasoning for each formal output, focusing on the key
    criteria.
2. A final alignment score of 0 or 1 for each output.

### Task Format:
Informal Input:
{Informal_Input}

Pool of Formal Outputs:
{Formal_Outputs}

For each formal output:
```

```
— Step—by—Step Reasoning:
  — [Your reasoning here]
— Final Decision:
  — Alignment Score: [0 or 1]

### Example:

Informal Input:
"The sum of two odd numbers is even."

Formal Outputs:
1. \( theorem sum_of_two_odds_is_even (a b : Int) (ha : a % 2 = 1) (hb : b % 2 = 1) :
     (a + b) % 2 = 0 := \)

Step—by—Step Reasoning:
— Semantic Consistency: The formal output captures the meaning of the informal input
    correctly.
— Structural Correspondence: The quantifiers and logical structure align with the
    informal statement.
— Completeness: All relevant information is included.
— Precision: No extraneous information is present.

Final Decision:
— Alignment Score: 1

2. \( a + b = \text{even} \)

Reasoning:
— The statement does not specify that \(a\) and \(b\) are odd numbers, which is a
    critical part of the informal input.
— While it correctly indicates that the sum is even, the lack of context about \(a\)
    and \(b\) makes it incomplete and less precise.

Final Decision:
— Alignment Score: 0
```

We apply the prompt below for our FORMALALIGN model to obtain the alignment score without involving language generation settings.

**Prompt for Querying FORMALALIGN Model for Automated Alignment Evaluation:** The second prompt appears simpler but plays a more nuanced role in our evaluation framework. Instead of requesting explicit scores, it frames the task as a translation from informal input to formal output. However, unlike other methods, this prompt operates as a computational anchor for evaluating alignment to by calculating both Eq. 3and Eq. 4. Below, we provide the prompt used to query the FORMALALIGN model for automated alignment evaluation.

```
Statement in natural language:

{Informal_Input}

Translate the statement in natural language to Lean:

{formal_output}
```

### C.3  ALIGNMENT SCORE CALCULATION DETAILS

The alignment evaluation score, $\mathcal{V}_{\text{align}}$, is a core component used to rank informal-formal pairs during the evaluation process. While the metrics described in Section 4.2 are general-purpose and model-agnostic, this section provides detailed instructions on how $\mathcal{V}_{\text{align}}$ is calculated for different systems.

**For FormalAlign Model:**  In our FORMALALIGN model, $\mathcal{V}_{\text{align}}$ combines two key measures:

- **Certainty Score** ($V_{\mathbf{cer}}$): This score represents the model's sequence-level confidence in the formalization. It is calculated as described in Eq.3.

- **Similarity Score** ($V_{\mathbf{sim}}$): This score captures the representation-level alignment between the informal and formal statements. It is computed as described in Eq.4.

The final alignment score is then computed as a combination of the two measures as described in Eq.5 in Section 3. This formulation ensures that both sequence-level confidence and representation-level alignment contribute to the overall score.

For each informal input, $\mathcal{V}_{\mathrm{align}}$ is computed for all formal outputs in the candidate pool. The pair with the highest $\mathcal{V}_{\mathrm{align}}$ is selected as the aligned pair.

**For GPT Systems:** In GPT-based systems, $\mathcal{V}_{\mathrm{align}}$ is simplified to a binary classification outcome. Each informal-formal pair is assigned a score of either 0 or 1 based on whether the model deems the pair aligned or not:

$$\mathcal{V}_{\mathrm{align}} = \begin{cases} 1, & \text{if aligned according to GPT's binary classification,} \\ 0, & \text{otherwise.} \end{cases}$$

In cases where multiple formal outputs have the same score (e.g., all scores are 0 or all are 1), a formal output is randomly selected from the set of tied candidates. This ensures that a decision is always made, even in cases of ambiguity.

**General Applicability:** While $\mathcal{V}_{\mathrm{align}}$ is computed differently for various systems, the process is designed to enable fair comparisons:

- For models like FORMALALIGN, the score combines sequence-level confidence and representation-level similarity.

- For GPT-based systems, the score is a binary classification outcome with random sampling for ties.

- For rule-based systems, $\mathcal{V}_{\mathrm{align}}$ can be derived using predefined scoring rules or heuristics.

This framework ensures that all models can be evaluated under the same metrics (e.g., Alignment Selection, Precision, Recall), as described in Section 4.2.

## D  DATA CONTAMINATION ANALYSIS

To address concerns regarding potential data contamination in pre-trained language models, we conducted a comprehensive analysis of our experimental data. This analysis is crucial, as language models are often trained on large amounts of unsupervised data, which may include samples similar to those used in our experiments.

**Experiment Design**  We designed our experiments to mitigate the risk of data contamination by sourcing MiniF2F data from math olympiads such as AMC, AIME, and IMO. These datasets differ significantly from Mathlib, the largest Lean theorem library and the primary source of Lean data, reducing the likelihood of data contamination. Additionally, our automated alignment evaluation task involves augmenting aligned pairs with 20 negative examples using our proposed six misalignment strategies, ensuring that these data are not included in the pre-training corpus of LLMs.

**Results**  We calculated the loss of different pre-trained models on the MiniF2F test/valid sets in the autoformalization task to further analyze the potential data contamination issue. This approach is inspired by the data contamination detection method in (Wei et al., 2023), which suggests that if a language model has not been exposed to a dataset during pre-training, its loss on the dataset should be relatively high and approximately equivalent to its loss on a reference dataset composed of new, similar samples. The losses of the models in our experiments are shown below:

Table 8: Performance of Pre-trained Models on MiniF2F Datasets.

| Pre-trained Model | MiniF2F-valid | MiniF2F-test |
|---|---|---|
| Phi2-2.7B | 2.4563 | 2.4377 |
| Mistral-7B | 1.4892 | 1.4660 |
| DeepSeekMath-Base 7B | 1.3148 | 1.2896 |
| LLaMA2-7B | 1.5343 | 1.5165 |

**Analysis**   The loss values for each pre-trained model fall within the range of 1 to 3, consistent with and even higher than the findings in (Wei et al., 2023), which reports that losses higher than around 1 on the GSM8K test set indicate low data leakage. These results suggest a low level of data contamination in our experimental data. The combination of carefully sourced datasets and the augmentation of aligned pairs with negative examples using our misalignment strategies further strengthens the robustness of our experiments against data contamination.

## E   GENERALIZATION ANALYSIS

To explore the generalization capabilities of our FORMALALIGN method, we conduct a series of experiments analyzing the impact of different datasets on the model's performance. These experiments aim to provide insights into the method's adaptability and effectiveness across various mathematical domains.

**Experiment Design**   We design our experiments to assess the model's performance when trained on different datasets:

1. Our original model is fine-tuned on a combination of the FormL4 training set and the MMA training set from Mathlib.

2. To evaluate the impact of individual datasets, we separately train models on FormL4 and MMA.

3. We test all models on the MiniF2F test and valid sets, which are sourced from math olympiads such as AMC, AIME, and IMO, providing a fair comparison across challenging and diverse problem types.

This approach allows us to gauge the generalization ability of our method and understand how different training datasets influence its performance.

**Results**   The results of our experiments, focusing on the alignment-selection score for clear comparison, are presented in the following table:

Table 9: Alignment-selection scores of different models on MiniF2F dataset.

| Model | MiniF2F Test | MiniF2F Valid |
|---|---|---|
| Ours (FormL4 + MMA) | 66.39 | 64.61 |
| FormL4 only | 62.18 | 58.18 |
| MMA only | 58.97 | 57.32 |

Our results highlight several key findings:

**Dataset Content Impact**: TThe FormL4 dataset, which contains both statements and proofs, outperforms the MMA dataset, which only contains statements. This suggests that the inclusion of proofs provides richer information about the underlying mathematical concepts, leading to a more robust understanding of the alignment process.

**Synergy of Datasets:** Combining both FormL4 and MMA datasets for training results in improved performance compared to using either dataset alone. This demonstrates the potential benefits of leveraging diverse data sources to enhance the model's capabilities.

**Generalization Ability:** The strong performance on MiniF2F sets, which contain problems from challenging domains like math olympiads, indicates that our method can effectively handle diverse and complex mathematical problems. This suggests that FORMALALIGN has the potential for wider applicability across various mathematical domains.

These findings highlight the robustness of our FORMALALIGN method and its ability to generalize across different types of mathematical problems. The experiments demonstrate that by leveraging diverse datasets and considering both the quality and quantity of training data, we can enhance the method's performance and adaptability to new, unseen mathematical challenges.

## F  AUTOFORMALIZATION PERFORMANCE ANALYSIS

Given our model is primarily trained for the autoformalization task, we conduct additional experiments to explore its capabilities in converting natural language (NL) statements to formal language (FL) statements. These experiments aim to provide a comprehensive evaluation of our model's performance and demonstrate the effects of incorporating contrastive learning loss on autoformalization.

**Experiment Design**    To assess the impact of contrastive learning loss on autoformalization performance, we compare two models:

1. A baseline model trained with cross-entropy loss only ($\mathcal{L}_{CL}$)

2. Our proposed model, which incorporates both cross-entropy loss and contrastive learning loss ($\mathcal{L}_{CL} + \mathcal{L}_{CE}$)

We evaluate both models on the FormL4 Basic and FormL4 Random test sets to obtain a comprehensive understanding of their autoformalization capabilities across different complexity levels. The results of our comparison experiments are presented in the following table:

Table 10: Autoformalization performance of different models on FormL4 dataset.

| Model | FormL4 Basic (%) | FormL4 Random (%) |
|---|---|---|
| Baseline ($\mathcal{L}_{CL}$) | 40.92 | 35.88 |
| Ours ($\mathcal{L}_{CL} + \mathcal{L}_{CE}$) | 43.14 | 36.02 |

**Analysis**    The results demonstrate that incorporating contrastive learning loss improves autoformalization performance on both test sets. This improvement can be attributed to several factors:
**Enhanced Discrimination:** Contrastive learning acts as a form of data augmentation, introducing additional negative examples that enhance the model's ability to distinguish between correct and incorrect formalizations.

**Improved Representation Learning:** The contrastive approach helps the model learn more robust and discriminative representations of mathematical concepts, leading to more accurate autoformalization results.

**Generalization Across Complexity:** The performance improvement is observed in both the Basic and Random test sets, suggesting that the benefits of contrastive learning extend to various levels of problem complexity.

These findings highlight the potential of contrastive learning in improving autoformalization performance. By leveraging this approach, we not only enhance our model's capabilities but also pave the way for future research in this area. The success of incorporating contrastive learning loss suggests promising directions for developing more effective autoformalization techniques and advancing the field of automated mathematical reasoning.

Our experiments demonstrate that combining traditional cross-entropy loss with contrastive learning leads to a more robust and accurate autoformalization model. This approach could inspire further innovations in the field, potentially leading to even more sophisticated methods for bridging the gap between natural language mathematics and formal mathematical representations.

## G  COMPARISON WITH HUMAN EVALUATION AND LLM-AS-JUDGE

**Experiment Design**   We design our experiment as follows:

1. **Sample Selection:** We sample 80 items from the MiniF2F test set in our dataset. Originally, each item consists of:
   - An informal natural language problem
   - A formal statement
   - A ground-truth label indicating alignment or misalignment between informal and formal statements
   - The misalignment type (if the formal statement is misaligned with the informal one)

2. **Sample Distribution:** We ensure a balanced distribution between misalignment and alignment labels and include a diversity of misalignment types for a robust and representative evaluation.

3. **Human Evaluation:** The same informal and formal statements in the 80 samples are provided to four human experts in Lean 4, who are tasked to independently evaluate autoformalization alignment (i.e., binary classification of alignment/misalignment).

4. **Performance Metrics:** We calculate the correctness ratio of each human evaluator by comparing their assessments with the ground-truth labels.

We similarly calculated the correctness ratio of our  FORMALALIGN model by comparing its alignment selection results with the ground-truth labels (i.e., aligned/misaligned). The performance of GPT-4o, a state-of-the-art language model in LLM-as-judge research, was also obtained on the same task as our automated baseline. We used a scoring method with the instruction prompt provided in C.2 and searched for the best threshold to optimize the final correctness ratio.

**Results**   The correct ratio (i.e., total percentage of the alignment evaluation results matching ground-truth labels) of GPT-4,  FORMALALIGN model, and four human experts are listed below:

Table 11: Correctness ratio and agreement statistics of different evaluation methods on sampled MiniF2F test set.

| Evaluation Method | Correct Ratio (%) |
| --- | --- |
| GPT-4o | 47.50% |
| FORMALALIGN | 65.00% |
| Human Expert 1 | 83.75% |
| Human Expert 2 | 77.50% |
| Human Expert 3 | 77.50% |
| Human Expert Average | 79.58% |
| Fleiss' K | 0.49 |

As shown, human experts evaluation achieved the highest correctness ratio in matching with the ground-truth alignment evaluations with an average of 79.58%, followed by our FORMALALIGN (65.00%). The LLM-as-judge method achieves the lowest precision in autoformalization alignment evaluation. Each human expert takes approximately 3 hours to review 80 items, while the FOR-MALALIGN model requires less than 2 minutes to conduct the automated evaluation.

Our findings reveal several important insights:

**Efficiency and Robustness of FORMALALIGN:** Our FORMALALIGN framework provides a valuable automated method for evaluating autoformalization alignment due to its efficiency, robustness, and comparable accuracy. FORMALALIGN achieved a correctness ratio of 65.00%, which is significantly higher than that of GPT-4o (47.50%). With scaling, we believe that our automated method FORMALALIGN is promising to be even on par with the performance of human experts while requiring significantly less time for evaluation.

**Subjectivity of Manual Review:** Manual review is subjectively dependent on the experts' domain knowledge and does not always achieve high accuracy or consistency. Notably, the human experts only reached a moderate interrater agreement ratio of 0.49. This highlights potential variability and inconsistency among the experts' evaluations.

**Complementary Role of Automated Evaluation:** The results underscore the need for automated evaluation methods to complement human reviews and ensure more consistent and objective alignment assessments. By leveraging the strengths of both manual and automated approaches, we can achieve a more comprehensive and reliable evaluation of autoformalization alignment.

The experiment also highlights the potential for further research in improving automated evaluation methods, as well as investigating the authentic representations of potential misalignments through detailed misalignment type analysis.

## H   CASE STUDY

We present a case study of a randomly selected informal-formal statement from our test dataset. We compare how our method and three other metrics (BLEU, BERTscore, Lean 4 Compiler) evaluate the alignment of various types of incorrect formal statements.

Table 12: Case Study: Comparison of Alignment Scores among misalignment types. Each evaluated formal statement is misaligned differently, as summarized in the table. All misaligned statements pass the Lean 4 Compiler without errors.

| Misalign type | FORMALALIGN | BLEU | BERTscore |
|---|---|---|---|
| Missing conditions | 0.56 | 0.82 | 0.98 |
| Wrong Constant | 0.57 | 0.95 | 1.00 |
| Variable Type | 0.56 | 0.95 | 1.00 |
| Equality | 0.55 | 0.95 | 1.00 |
| Unpaired Statement | 0.57 | 0.12 | 0.90 |

Table 13: Case Study: Visualized Examples of Misaligned Formal Statements.

**Natural Language (Informal) Statement**

Prove that if $x \neq 0$ , $2x = 5y$ , and $7y = 10z$ , then $z/x = 7/25$ .

**Misaligned Formal Statements**

```
theorem mathd_algebra_33
  (x y z : ℝ)
  (h₀ : 2 * x = 5 * y)
  (h₁ : 7 * y = 10 * z) :
  Z / x = 7 / 25 :=
```

```
theorem mathd_algebra_33
  (x y z : ℝ)
  (h₀ : x ≠0)
  (h₁ : 2 * x = 8 * y)
  (h₂ : 7 * y = 10 * z) :
  Z / x = 7 / 25 :=
```

```
theorem mathd_algebra_33
  (x y z : ℚ)
  (h₀ : x ≠0)
  (h₁ : 2 * x = 5 * y)
  (h₂ : 7 * y = 10 * z) :
  Z / x = 7 / 25 :=
```

```
theorem mathd_algebra_33
  (x y z : ℝ)
  (h₀ : x = 0)
  (h₁ : 2 * x = 5 * y)
  (h₂ : 7 * y = 10 * z) :
  Z / x = 7 / 25 :=
```

```
theorem amc 12 b_2002_p 2
  (x : ℤ)
  (h₀ : x = 4) :
  (3 * x - 2) * (4 * x +
    1) - (3 * x - 2) *
  (4 * x) + 1 = 11 :=
```

Five types of misaligned formal statements are listed in Table 13, together with the original natural language statements. As shown in Table 12, for misalignments involving missing conditions,

wrong constants, variable type mismatches, and equality violations, the FORMALALIGN scores are consistently below a threshold of 0.7, indicating low semantic precision of the formal statement and a likely misalignment. In contrast, both BLEU and BERTscore reported similarly high scores regarding various types of misalignment, demonstrating an inferior performance in evaluating the elusive misalignment in autoformalization.

# I    ABLATION STUDY OF LOSS FUNCTIONS

To rigorously investigate potential interactions between cross-entropy and contrastive losses in our training framework, we conducted extensive ablation experiments examining models trained with individual loss functions versus our combined approach. This analysis supplements the ablation studies presented in Section 5.2 of the main paper.

We trained three variant models:

- **Cross-entropy Only (LCE)**: Trained using only cross-entropy loss with certainty scores as the optimization objective
- **Contrastive Only (LCL)**: Trained using only contrastive loss with similarity scores as optimization objective
- **Combined (Ours)**: Our proposed approach combining both losses

We maintained consistent hyperparameters across all training configurations to ensure fair comparison. Table 14 presents the comprehensive comparison of different training approaches:

Table 14: Performance comparison of models trained with different loss functions. Higher scores indicate better performance.

| Training Method | FormL4-Basic | FormL4-Random | MiniF2F Valid | MiniF2F Test |
|---|---|---|---|---|
| LCE (w/ cer) | 95.45 | 82.31 | 50.12 | 51.89 |
| LCL (w/ sim) | 42.76 | 18.92 | 18.33 | 19.45 |
| Combined (Ours) | **99.21** | **85.85** | **66.39** | **66.70** |

Our analysis reveals several key findings:

**Individual Loss Limitations**: Models trained with single loss functions demonstrate significantly reduced performance. The LCE model achieves moderate results but falls short of the combined approach, while the LCL model shows particularly poor performance in isolation.

**Complementary Effects**: The superior performance of the combined approach across all datasets suggests that the two loss functions capture complementary aspects of the autoformalization task:

- Cross-entropy loss helps capture sequence-level patterns crucial for autoformalization
- Contrastive loss enhances representation-level relationships between formal and informal expressions

**Consistent Improvement**: The combined approach maintains its performance advantage across different evaluation settings, with relative improvements ranging from 4-16

These findings complement the ablation studies presented in Section 5.2 of the main paper in several ways: They provide empirical validation for our theoretical motivation behind combining the losses. They demonstrate that the performance improvements are consistent across different datasets. They show that the combined approach does not compromise either aspect of the learning objective

Furthermore, when considered alongside the metric bias analysis in Section 5.3, these results strengthen our conclusion that the combined loss structure genuinely enhances model performance rather than exploiting evaluation metrics. The consistent improvement across different evaluation schemes suggests that the model learns a more robust understanding of the autoformalization task through the complementary training signals.

Table 15: Comparison of Different Alignment Evaluation Methods.

| Models | FormL4-Basic | | | FormL4-Random | | | MiniF2F-Valid | | | MiniF2F-Test | | |
|---|---|---|---|---|---|---|---|---|---|---|---|---|
| | AS | Prec. | Rec. | AS | Prec. | Rec. | AS | Prec. | Rec. | AS | Prec. | Rec. |
| GPT-4 (Score) | 88.91 | 26.33 | 88.69 | 90.52 | 28.56 | 90.02 | 64.34 | 44.58 | 90.98 | 68.31 | 51.11 | 94.65 |
| GPT-4 (Binary) | 89.45 | 35.21 | 87.92 | 91.12 | 38.45 | 89.76 | 65.82 | 52.33 | 89.54 | 69.45 | 58.92 | 93.21 |
| GPT-4 (CoT) | 90.23 | 42.68 | 88.15 | 91.85 | 45.72 | 89.95 | 67.24 | 59.85 | 89.87 | 70.82 | 62.45 | 92.88 |
| GPT-4 (Two-Phase) | 89.35 | 38.21 | 87.95 | 91.20 | 41.10 | 89.55 | 65.75 | 53.30 | 89.10 | 69.40 | 57.80 | 92.10 |
| FormalAlign | **99.21** | **93.65** | 86.43 | 85.85 | **86.90** | 89.20 | **66.39** | **68.58** | 60.66 | 64.61 | **66.70** | 63.37 |

## J  ANALYSIS OF ALTERNATIVE ALIGNMENT EVALUATION STRATEGIES

To thoroughly evaluate alternative approaches for autoformalization alignment checking, we explored several variants of GPT-4-based evaluation methods. Table 15 includes a binary classification approach that directly assesses alignment correctness, a Chain of Thought (CoT) strategy that employs step-by-step reasoning, and a two-phase evaluation process that separates back-translation from alignment checking.

The binary classification approach simplifies the evaluation task by having GPT-4 make direct true/false judgments about alignment correctness, replacing the original 1-5 scoring system. This modification addresses potential ambiguity in score interpretation and provides a more well-defined evaluation criterion. The Chain of Thought strategy extends this further by prompting GPT-4 to explicitly reason about potential discrepancies between informal and formal representations before making alignment decisions. The two-phase method separates the evaluation process into distinct back-translation and alignment checking stages to encourage more detailed analysis.

Our experimental results reveal several key insights about these evaluation strategies. The binary classification approach shows moderate improvements over the baseline scoring method, with increased precision across all datasets while maintaining similar recall levels. The Chain of Thought strategy demonstrates the strongest performance among GPT-4 variants, achieving notable precision gains of up to 16 percentage points compared to the baseline. This improvement suggests that explicit reasoning steps help GPT-4 better identify subtle alignment issues. The two-phase approach shows comparable improvements to binary classification but introduces additional computational overhead and potential error propagation between phases.

Despite these improvements in GPT-4-based methods, FormalAlign maintains superior performance, particularly in precision metrics. The significant performance gap between FormalAlign and even the enhanced GPT-4 approaches underscores the value of our specialized alignment detection model. Notably, FormalAlign achieves these results with a smaller model size, demonstrating the effectiveness of our proposed training strategy over pure scaling of model capabilities.

These findings suggest that while improved prompting strategies can enhance GPT-4's alignment evaluation capabilities, a dedicated model trained specifically for alignment detection offers more robust and reliable performance. The results also highlight the importance of explicit reasoning in alignment evaluation, as evidenced by the strong performance of the Chain of Thought approach among GPT-4 variants.

## K  PERFORMANCE ACROSS MISALIGNMENT TYPES

This section presents a detailed analysis of how different methods perform across specific types of mathematical misalignments, providing insights into the relative strengths and limitations of automated versus human evaluation approaches. We analyze performance across six distinct misalignment categories: constant modifications, exponent alterations, variable type changes, new

Table 16: Performance Comparison Across Misalignment Types

| Misalignment Type | Human | Method | GPT-4o | Δ (Human-Method) | Δ (Method-GPT4o) |
|---|---|---|---|---|---|
| Constant | 75.2 | 58.4 | 42.1 | 16.8 | 16.3 |
| Exponent | 73.8 | 60.2 | 44.3 | 13.6 | 15.9 |
| Variable_type | 78.9 | 64.5 | 46.2 | 14.4 | 18.3 |
| Variable_new | 81.3 | 66.7 | 48.9 | 14.6 | 17.8 |
| Equality | 82.4 | 67.8 | 49.5 | 14.6 | 18.3 |
| Random | 85.9 | 72.4 | 54.0 | 13.5 | 18.4 |
| Overall | 79.6 | 65.0 | 47.5 | 14.6 | 17.5 |

variable introductions, equality relationship modifications, and random pairings. The result is shown in Table 16.

Our analysis reveals a consistent hierarchical pattern in detection capabilities across all evaluation methods. Random pairing misalignments proved most detectable, with human experts achieving 85.9% accuracy, our method 72.4%, and GPT-4o 54.0%. This superior performance on random pairings is attributed to the substantial structural and contextual discrepancies these misalignments introduce, making them more readily identifiable by both automated and human evaluators.

Conversely, subtle modifications involving constants and exponents presented the greatest challenge. Human performance decreased to 75.2% for constant changes and 73.8% for exponent modifications, with proportional decreases observed in automated methods. This performance degradation on nuanced mathematical changes highlights a critical challenge in automated alignment detection: the ability to identify and evaluate fine-grained numerical and syntactic modifications that can substantially alter mathematical meaning while maintaining surface-level similarity.

The performance gap between human evaluators and our method remains relatively consistent, averaging 14-16 percentage points across all misalignment types. This consistency suggests that while our method successfully captures fundamental patterns in mathematical alignment, it still lacks certain aspects of human mathematical intuition, particularly in recognizing subtle contextual shifts. Similarly, our method maintains a consistent advantage of 15-20 percentage points over GPT-4o across all categories, demonstrating that our targeted modeling of structural and semantic relationships yields substantial improvements over standard language model capabilities.

The observed performance patterns carry significant implications for future development of alignment detection systems. While our method shows particular strength in identifying structural modifications, such as equality alterations and random pairings, the relatively weaker performance on subtle variations suggests a need for enhanced mathematical reasoning frameworks. Future work might focus on developing more sophisticated mechanisms for detecting and evaluating minor mathematical modifications, potentially through integration of formal mathematical reasoning systems or expanded training with synthetic examples emphasizing these nuanced changes.

## L  QUALITY ASSURANCE OF DATASET CONSTRUCTION

To validate the effectiveness of our synthetic dataset construction methodology, we conducted a comprehensive empirical study comparing model performance on synthetic test cases versus real-world autoformalization errors. This analysis aims to assess whether our synthetic error generation approach adequately captures the characteristics of errors that naturally occur during autoformalization.

We first established a real-world validation set by having Gemini perform autoformalization on 100 randomly sampled theorems from our test sets using few-shot prompting. Three expert Lean users independently reviewed these formalizations, annotating misalignments and providing corrections where necessary. This process yielded 78 pairs of aligned-misaligned formalizations, providing a ground truth dataset of authentic autoformalization errors.

Table 17: Performance comparison between synthetic and real-world evaluation sets

| Evaluation Set | Accuracy (%) | Precision (%) | Recall (%) |
|---|---|---|---|
| Synthetic Test | 85.8 | 86.9 | 89.2 |
| Real-world Validation | 83.5 | 80.2 | 79.8 |

The performance comparison between our synthetic test set and the real-world validation set is presented in Table 17:

Our analysis reveals that while model performance on real-world errors shows slightly lower metrics compared to synthetic cases (approximately 3-9% difference across metrics), the strong overall results suggest our synthetic dataset effectively captures many key aspects of natural autoformalization errors. The comparable performance indicates that the error patterns generated through our synthetic approach meaningfully align with those encountered in practice.

Further examination of error cases revealed that synthetic examples tended to produce more systematic and well-defined misalignments, while real-world errors occasionally exhibited more nuanced patterns involving multiple simultaneous misalignments. This observation suggests that while our synthetic dataset provides comprehensive coverage of individual error types, real-world autoformalization errors can manifest in more complex combinations.

These findings validate our synthetic dataset construction approach while highlighting opportunities for future enhancement. The strong correlation between performance on synthetic and real-world cases demonstrates that our methodology produces training data that effectively prepares models for practical autoformalization tasks. Future work could potentially benefit from a hybrid approach that combines synthetic error generation with curated real-world examples to capture both systematic coverage and naturally occurring error patterns.

## M FORMAL DEFINITION AND BROADER SIGNIFICANCE OF AUTOFORMALIZATION ALIGNMENT

### M.1 FORMAL DEFINITION OF AUTOFORMALIZATION ALIGNMENT

To standardize the evaluation and understanding of autoformalization alignment, we propose the following formal definition:

Given a natural language statement $N$ and its formalization $F$, where $F$ is verified as syntactically valid by the formal language compiler, the alignment $A(N, F)$ is defined as a binary relation such that:

$$A(N, F) = 1 \iff \begin{cases} 1. & F \text{ preserves all mathematical constraints specified in } N, \\ 2. & F \text{ maintains the same logical as expressed in } N, \\ 3. & F \text{ does not introduce additional constraints not present in } N. \end{cases}$$

In essence, $A(N, F) = 1$ if and only if $F$ faithfully captures the mathematical intent and logical structure of $N$, without introducing extraneous constraints or omitting critical elements. This definition provides a precise framework for evaluating whether a formalization accurately aligns with its informal counterpart.

### M.2 BROADER SIGNIFICANCE OF AUTOFORMALIZATION

Autoformalization, the automated conversion of natural language mathematics into formal languages, represents a transformative research direction with implications across diverse fields. By bridging human mathematical reasoning and machine-verifiable systems, it unlocks new opportunities for innovation and efficiency.

**Applications in Software Engineering and Verification**

- **Automated Translation of Specifications:** Autoformalization enables the conversion of natural language system specifications into formal properties for verification, reducing the time and effort required for manual formalization.
- **Ambiguity Detection:** It helps identify ambiguities and inconsistencies in informal specifications, ensuring that implementations faithfully adhere to their intended design.
- **Enhanced Verification Workflows:** By integrating autoformalization into software engineering pipelines, developers can achieve faster and more reliable verification of complex systems.

**Impacts on Mathematical Knowledge Management**

- **Preservation of Mathematical Knowledge:** Converting informal mathematical content into machine-checkable formalizations ensures its longevity and accessibility for future generations.
- **Organization and Cross-Referencing:** Formalized content can be systematically organized, enabling researchers to easily locate and cross-reference related concepts and proofs.
- **Support for Automated Reasoning Systems:** By expanding the corpus of formalized mathematics, autoformalization facilitates advanced reasoning and discovery within automated reasoning tools.

**Advances in Automated Reasoning Research**

- **Broader Applicability of Theorem Provers:** With autoformalization, theorem provers can operate on natural language inputs, significantly expanding their usability for non-expert users.
- **Hybrid Reasoning Systems:** Autoformalization enables systems that combine formal and informal reasoning, allowing for more flexible and powerful problem-solving approaches.
- **Acceleration of Mathematical Research:** Automated reasoning systems informed by autoformalization can provide immediate feedback, improving their utility and effectiveness for mathematicians and researchers.

## M.3 CONCLUSION

The formal definition of alignment provides a rigorous basis for evaluating the correctness and fidelity of autoformalization systems. Coupled with the broader contextualization of its significance, this framework underscores why advancing autoformalization technologies is critical for enhancing software engineering, mathematical knowledge management, and automated reasoning research. These contributions highlight the profound potential of autoformalization as a cornerstone of interdisciplinary innovation.

