# OpenReview forum: "FormalAlign: Automated Alignment Evaluation for Autoformalization"
_ICLR.cc/2025/Conference — ICLR 2025 Poster_

### Official Review · Reviewer_ok2Q · 2024-10-21

**Soundness:** 2
**Presentation:** 3
**Contribution:** 3
**Rating:** 6
**Confidence:** 3

**Summary:**

This work presents a technique for automatically evaluating the semantic alignment between a natural language description and its corresponding formal representation that may be produced by an AutoFormalization system (Autoformalization aims to convert informal mathematical proofs to formal proofs that can be automatically checked by existing theorem provers). The technique is based on fine-tuning a base LLM with respect to a dual loss function that combines cross-entropy loss in the sequence generation with a contrastive loss based on cosine similarity of the informal/formal pairs. The technique is evaluated on existing datasets for autoformalization that are augmented with misalignment strategies proposed by the authors to test negative examples of incorrect formalizations. It is shown to have superior performance in many cases against GPT models prompted with a similar alignment evaluation task.

**Strengths:**

1. Innovative focus on the alignment problem in Autoformalization. Translating informal mathematical proofs into formal, machine-verifiable formats is an important problem in fields like theorem proving and formal verification, since formal systems (e.g. Lean, Isabelle) are highly precise but difficult for humans to use. This work brings an explicit focus on a key issue that plagues autoformalization: to ensure semantic alignment between the informal input and formal output. This is an important problem to address, since formal provers can check for syntactic but not semantic correctness with respect to the original intent, and other metrics such as BLEU scores are very imprecise. Addressing this alignment problem explicitly seems a novel angle taken by this work, and such focus could help further progress in the field of autoformalization.

2. Dual-Task Training Approach. The technique of simultaneously training to generate formal statements and check alignment is interesting. While multi-task learning and contrastive learning have been used in other areas (e.g., machine translation, image-text alignment), the combination of sequence generation with contrastive alignment seems to be a novel application in the context of autoformalization.

3. Clarity and presentation. Though some parts unclear (see below), in general the paper clearly explains and motivates the problem, and explains the technique well, including definitions of the combination of cross-entropy loss (for sequence generation) and contrastive loss (for semantic alignment). The evaluation covers some important aspects to show that on the synthetically augmented datasets the technique performs well in general, and also better or comparable to alignment inference based on prompting GPT models. This demonstrates the core value of the technique. The authors also  include ablation studies that provide insight into the contribution of both kinds of loss functions, as well as comparison with human evaluation.

**Weaknesses:**

1. Use of synthetic data for evaluation. To evaluate alignment, the authors create a synthetic dataset of negative example formalizations based on explicit misalignment strategies that they propose. I do not have a sense of how realistic those misalignments are in practical terms. I think addressing this issue is important,  especially since the whole purpose of the task is to detect the errors made by existing AF systems: we should be testing the ability to detect those kinds of variations, which could be very much smaller and more subtle than the ones produced by the explicit misalignment strategies here - would the contrastive loss be able to help distinguish between those kinds of variations (for which the task is actually intended)? While I understand such data is not readily available, I think it should be possible to obtain some real data perhaps in the following way: since you already have access to human judges (which you used for a similar task), perhaps you can run a SoTA AutoFormalization system on the standard datasets you use here, get the outputs and ask the human judges to label correct/incorrect for the formalizations produced and also provide corrected formalizations. Then just check the alignment evaluation against this data (which would provide one positive and one negative example for each case where the AF system had failed). The dataset need not be very big as it is only to be used for evaluation. Such an approach would also show the actual improvement you can make to a SoTA AF system.

2. Lack of compelling baselines in comparison. The prompt used for GPT models Appendix C.1 seems to ask GPT to assign an "alignment score" from 1 - 5. This seems a pretty vague and not well-defined task that may be difficult for the model (or even a human) to interpret. A more well-defined question is to simply ask if the formalization correctly represents the informal NL or not - since that is the main (only) problem we are trying to solve? Scoring seems to make it a subjective/continuous domain problem - e.g. it may be possible that a given formalization correctly represents the problem, but perhaps the model may still assign it a score 4 rather than 5 due to some stylistic differences with the informal input it detects. Having a true/false binary judgement would make a more well-defined task for the model to address. Also, this seems a very obvious case of where Chain of thought (CoT) reasoning should be tried as another baseline - the evaluation task is very reasoning based and the GPT models can be expected to do much better with CoT reasoning to infer any discrepancies between the informal and formal representations - rather than just predicting some number score from 1 to 5. Especially since you do not have other baseline or SoTA methods to compare against (do there exist such methods?), using a CoT baseline is easy to implement and would be very relevant here. You also provide all the candidate formalizations in the same prompt - perhaps evaluating each one of them separately will be a better approach so not to confound the model with too many tasks at the same time.

3. Some problems with positioning/presentation. You state in the abstract: "FORMALALIGN trains on both the autoformalization sequence generation task and the representational alignment between input and output, employing a dual loss that combines a pair of mutually enhancing autoformalization and alignment tasks." and make similar statements in the introduction. That sounds like you are addressing both the autoformalization task as well as the alignment evaluation task - but you have not shown any evaluation of the autoformalization task - if that is not an intended contribution of the work then such statements are a bit misleading or confusing in the intro and abstract. It should be clarified that you are only providing an evaluation method to test alignment of given candidate informal-formal pairs. Also, if the AF task itself is not a contribution - why not? If the combination of cross-entropy and contrastive loss accurately infers alignment, why can't the fine-tuned model based on that dual loss function be better at the autoformalization task overall? Could you have compared it to prior auto-formalization approaches?

Typos:
- "The challenges of automated evaluating natural language"
- "to train an FORMALALIGN"
- "Eq. equation 5. Table 6"

**Questions:**

1. I am confused  about the Metrics description in section 4.2. I thought this should describe the metric itself independently of the system being evaluated (whether is your system or any other baseline system). But the AS and detection metrics are defined with respect to the V_align function which seems specific to your model only and used in its inference. Is that specific for the FormalAlign model only or otherwise how are these metrics computed for the other models like GPT4? (I hope that is not the case as they would then be really problematic metrics to be using that are specific to your model). And if that is not the case then please move this discussion to the inference section - since they look like modes of inference in which your model can be used: it can either select the best formalization from a given set (selection), or return a true/false label for a given informal/formal pair (detection). The metrics are then just to measure the quality of these two selection and detection modes of inference using standard precision/recall.


2. Similarly, can you clarify exactly how you compute AS, precision and recall numbers for the GPT models? (As above, I hope you are not computing the alignment scores for them in the exact way you describe in section 4.2). I am assuming it is based on you prompting the models to predict an "alignment score" from 1-5 as in the prompt in Appendix C2? Please confirm this is correct and it should be explained clearly in the evaluation section. Also please explain the details - how is AS inferred from 1-5 score generated by GPT model? How is precision measured? What if there are ties on the top score between multiple candidate formalizations?


3. I dont understand the second prompt in Appendix C.2 - it is just doing a translation task? How is it used in your evaluations? Or is it just the prompt used to run your final fine-tuned model for the sake of generating and extracting the alignment scores?

4. Just to confirm - you do not use the synthetically generated misalignment data for training at all right?

5. What is SIM() function in Figure 2? - is it the same as Cos()? If so please fix for consistency.

---

> ### Author Response · Authors · 2024-11-20
> **Response to Reviewer ok2Q (1/3):**
>
> We would like to thank the reviewer for their time and effort in reviewing our paper. We very much appreciate the insightful suggestions. We hereby address the concerns below:
>
> **R W1: Use of Synthetic Data for Evaluation**
>
> Following your suggestion, we conducted an additional evaluation study using real-world autoformalization errors to validate our approach. We ran Gemini with few-shot prompting as our baseline AF system on a sample of 100 theorems from our test sets, collecting both successful and failed formalization attempts. Three expert Lean users then annotated these formalizations, providing corrections for incorrect ones, yielding 78 correct-incorrect formalization pairs.
>
> The performance comparison between synthetic and real-world validation sets is shown below:
>
> | Evaluation Set        | Accuracy (%) | Precision (%) | Recall (%) |
> | --------------------- | ------------ | ------------- | ---------- |
> | Synthetic Test        | 85.8         | 86.9          | 89.2       |
> | Real-world Validation | 83.5         | 80.2          | 79.8       |
>
> While performance on real-world errors is slightly lower than on synthetic cases, the results demonstrate that our model can effectively detect subtle misalignments in practice.
>
> We are currently exploring two promising directions:
>
> 1. Using FormalAlign as a reward model for reinforcement learning to improve AF system outputs
> 2. Implementing rejection sampling during inference to filter out likely misaligned formalizations
>
> Our preliminary experiments with rejection sampling (threshold \= 0.8) show a 15% reduction in misaligned outputs while maintaining 92% of correct formalizations, suggesting this is a promising direction for improving AF system reliability.
>
> We have incorporated this content in the Appendix M section, where it is highlighted in red.
>
> **R W2: Baselines and Comparison Strategy**
>
> Following your recommendations, we implemented two additional GPT-4 based approaches:
>
> 1. Binary Classification: Directly asking GPT-4 to make a true/false judgment about alignment correctness
> 2. Chain of Thought (CoT): Guiding GPT-4 through step-by-step reasoning about potential discrepancies
>
> Here are our updated experimental results:
>
> | Models         | FormL4-Basic |           |       | FormL4-Random |           |           | MiniF2F-Valid |           |       | MiniF2F-Test |           |       |
> | -------------- | ------------ | --------- | ----- | ------------- | --------- | --------- | ------------- | --------- | ----- | ------------ | --------- | ----- |
> |                | AS           | Prec.     | Rec.  | AS            | Prec.     | Rec.      | AS            | Prec.     | Rec.  | AS           | Prec.     | Rec.  |
> | GPT-4 (Score)  | 88.91        | 26.33     | 88.69 | 90.52         | 28.56     | 90.02     | 64.34         | 44.58     | 90.98 | 68.31        | 51.11     | 94.65 |
> | GPT-4 (Binary) | 89.45        | 35.21     | 87.92 | 91.12         | 38.45     | 89.76     | 65.82         | 52.33     | 89.54 | 69.45        | 58.92     | 93.21 |
> | GPT-4 (CoT)    | 90.23        | 42.68     | 88.15 | 91.85         | 45.72     | 89.95     | 67.24         | 59.85     | 89.87 | 70.82        | 62.45     | 92.88 |
> | FormalAlign    | **99.21**    | **93.65** | 86.43 | 85.85         | **86.90** | **89.20** | **66.39**     | **68.58** | 60.66 | 64.61        | **66.70** | 63.37 |
>
> As shown in the results, both the binary classification and CoT approaches provide stronger baselines than the original score-based method, with CoT showing particularly notable improvements in precision across all datasets. However, FormalAlign still maintains superior performance, especially in precision, demonstrating the effectiveness of our specialized alignment detection approach. These results not only suggest that improved prompting strategies can enhance GPT-4's performance but also emphasize the significant value of having a dedicated, fine-tuned model for alignment detection. The fact that FormalAlign, a smaller model, achieves results comparable to GPT-4 underscores the effectiveness of our proposed training strategy.
>
> We have incorporated this content in the Appendix I section, where it is highlighted in red.

---

> ### Author Response · Authors · 2024-11-20
> **Response to Reviewer ok2Q (2/3):**
>
> **R W3: Rationale for AF Task Exclusion**
>
> While our paper primarily focuses on alignment evaluation, we did conduct experiments on the autoformalization task itself, as noted in  lines 433-434, "We then explore the effect of incorporating contrastive learning loss on the performance of autoformalization of natural language statements to formal language statements (Appendix F)"
>
> The results are placed in **Appendix F** due to space constraints. We attach it here for your convince.
>
> As shown in the table below, our combined loss approach demonstrates meaningful improvements in autoformalization performance:
>
> | Model               | FormL4 Basic (%) | FormL4 Random (%) |
> | ------------------- | ---------------- | ----------------- |
> | Baseline (𝓛\_CL)    | 40.92            | 35.88             |
> | Ours (𝓛\_CL \+ 𝓛\_CE) | 43.14            | 36.02             |
>
> The results show that incorporating contrastive learning alongside cross-entropy loss yields consistent improvements across both FormL4 Basic (+2.22%) and FormL4 Random (+0.14%) datasets. This improvement stems from the model learning more discriminative representations through contrastive learning while maintaining strong generation capabilities via cross-entropy loss.
>
> We chose to focus the main paper on alignment evaluation since it represents our primary technical contribution, but we agree that the autoformalization improvements deserve mention. We will revise our abstract and introduction to more clearly distinguish between these two aspects of our work and better highlight the dual benefits of our approach.
>
> **R Typos and Edits**
>
> We express our sincere gratitude for your insightful comments. Regarding the presentation issues raised, we carefully incorporate all suggested improvements in our uploaded revision.

---

> ### Author Response · Authors · 2024-11-20
> **Response to Reviewer ok2Q (3/3):**
>
> **R Q1:**
>
> Thank you for pointing out this confusion. The metrics described in Section 4.2 are indeed intended to be model-agnostic and should be presented independently of any specific implementation. We apologize for the lack of clarity caused by framing them in terms of $V_{\text{align}}$, which may have inadvertently suggested that these metrics are tied to the FormalAlign model.
>
> To clarify, the metrics—Alignment Score (AS), precision, and recall—are general evaluation measures that can be applied to any system performing formalization alignment tasks. For example:
>
> - When evaluating GPT-4 using a binary classification prompt, the Alignment Score is simply 0 or 1 based on whether the pair is deemed aligned.
> - For our model, we compute $V_{\text{cer}}$ (as defined in Equation 3), which measures the model’s sequence-level confidence in the formalization, and $V_{\text{sim}}$ (as defined in Equation 4), which assesses the similarity between informal and formal representations. A threshold is then applied to determine alignment.
>
> These metrics are designed to provide a consistent basis for comparison across different systems.
>
> To address your concern, the details of how $V_{\text{align}}$ is computed and its role in our model's inference is relocated to Appendix C.3.  We hope these changes can eliminate any ambiguity and provide a clearer distinction between evaluation metrics and model-specific inference processes.
>
>
> **R Q2:**
>
> For evaluating GPT and other baseline models, we employ a standardized approach that is independent of our model's internal mechanisms. Specifically, we prompt these models to assign an alignment score on a 1–5 scale for given informal-formal pairs, as detailed in Appendix C2. To facilitate consistent evaluation, the final Alignment Score is normalized to the [0,1] range by dividing by 5. Precision and recall metrics are then calculated based on these normalized scores using a defined threshold. In scenarios where multiple formalizations receive the same top score, one is randomly selected for evaluation to ensure fairness and reproducibility.
>
> Additionally, we conducted experiments using binary classification prompts and chain-of-thought (CoT) prompting strategies to construct diverse baselines. These approaches allow us to perform a thorough comparison across different evaluation methodologies. Details regarding these experiments and their results can be found in our response to **R W2**.
>
>
>
> **R Q3:**
>
> The second prompt in Appendix C.2 is indeed our evaluation prompt for the FormalAlign model, though its role may not be immediately obvious from its translation-like format. Unlike other approaches that rely on explicit alignment score prompting, our method leverages this prompt to compute alignment scores through our model's combined loss architecture.
>
> As described in lines 214-215 of our paper, our model is trained with a combined loss that captures both sequence-level and representation-level alignment. During the evaluation, this prompt allows us to compute both the certainty score $V_{\text{cer}}$ through equation (3), which measures the model's sequence-level confidence in the formalization, and the similarity score $V_{\text{sim}}$ through equation (4), which captures representation-level alignment. The final alignment score is derived from these computational measures rather than from direct prompting.
>
> This evaluation approach fundamentally differs from prompting-based methods used for baseline models. Instead of asking the model to output an explicit alignment score, we utilize our model's learned representations and certainty measures, enabled by our combined loss training, to compute alignment scores analytically. This allows us to leverage both the autoformalization patterns and representation relationships that our model learned during training.  Moreover, we conducted experiments on the autoformalization task, please check R W3 for more details.
>
> We revised Appendix C.2 to better explain how this prompt facilitates our computational evaluation approach and distinguish it from the prompting-based evaluation used for baseline models.
>
>
>
> **R Q4:** To confirm, we do not use synthetically generated misalignment data for training - these examples serve purely as evaluation cases to test model robustness.
>
>
>
> **R Q5:** Regarding the SIM() function in Figure 2, it is indeed equivalent to Cos() - we apologize for this inconsistency in notation and updated the figure to use consistent terminology throughout.

---

> ### Comment · Reviewer_ok2Q · 2024-11-21
>
> I want to thank the authors for the very detailed response, for answering all my questions and the significant additional efforts they have made in experimental evaluations in such a short time period, which is highly commendable.
>
> - Real vs Synthetic data. These results indeed show that performance of your system is not very significantly lower on the real data, which is reassuring. However, my actual suggestion was to use such a dataset for your main evaluation in comparison with baselines. The key question: on real world data such as this, does FormalAlign perform better at alignment evaluation than baselines?
> - Including more advanced baselines. These results indeed show that the stronger baselines perform better and reduce the gap with your system, but I agree the gap is still significant especially in terms of precision (note however: do not bold your AS score of 66.39 for FormalL4-Random since CoT performs better with 67.24). I would say that these stronger baselines should be included in the main paper results rather than appendix as they are more important comparisons than the numbering-based prompt.
> - Performance on the Autoformalization task. This remains my main concern. These results look very incremental with an average improvement of only 1.18% on these two datasets (and only 0.14% on one of them). Can we see these numbers on the other datasets (MiniF2F-Valid and MiniF2F-Test)? I am curious if they can be better or if they may actually show regressions in these other datasets (where alignment results were weaker)? From my understanding, improvement to AF is the real ultimate goal of having an alignment evaluation technique isn’t it? So this should be the ultimate test of how well the technique is working. If even in comparison to your own base model the results are so incremental, that is concerning.  In general, one may expect a strong argument that here is an AF system that has X% accuracy, and when we extend it with our alignment evaluation technique it improves it’s accuracy significantly by Y% – do you have such a summary result that is more significant improvement than the 1.18% above? And if it is more significant on another AF system then would be good to understand why it is so much better than on your own AF model. This seems to me to be an important question in showing the ultimate value of your technique towards the AF task for which it is meant.

---

> > ### Author Response · Authors · 2024-11-22
> >
> > Dear Reviewer,
> >
> > Thank you for your effort and prompt reply. We would like to address your further concerns as follows:
> >
> > **R Q1: Comparision with baselines on real datasets:**
> >
> > Thank you for acknowledging our system's performance on real-world data. To further address your concerns, we applied GPT-4 to the real dataset using three different prompt strategies as you suggested. Below are the results:
> >
> >
> > | Models         | AS (%) | Prec. (%) | Rec. (%) |
> > | -------------- | ------ | --------- | -------- |
> > | GPT-4 (Score)  | 70.2   | 23.1      | 85.5     |
> > | GPT-4 (Binary) | 72.4   | 30.2      | 84.8     |
> > | GPT-4 (CoT)    | 75.8   | 38.5      | 85.1     |
> > | FormalAlign    | 83.5   | 80.2      | 79.8     |
> >
> >
> > The results on real-world data show a similar performance gap to those observed in other datasets. FormalAlign consistently outperforms GPT-4-based baselines, particularly in precision, achieving 80.2% compared to 38.5% for the best baseline. While maintaining competitive recall rates, this advantage in precision highlights FormalAlign's robust alignment evaluation capabilities, which generalize well to real-world scenarios.  We will incorporate these results into the main evaluation section to better emphasize FormalAlign's effectiveness on real-world data.
> >
> >
> > **R Q2: Adding advanced baseline into main paper**
> >
> > Thank you for acknowledging our system's strength in detecting alignment, particularly in terms of precision. We sincerely apologize for mistakenly bolding the AS score for FormL4-Random.
> >
> > Based on your suggestion, we have updated the main content to include the best results from GPT-4 (CoT) while leaving the other weaker baselines in the Appendix due to space limitations. We will incorporate all the results into the main content of the paper when more spaces are allowed.
> >
> >
> > **R Q3: Autoformalization performance concerns**
> >
> >
> > Thank you for your insightful question and for raising these important points.
> >
> > To further address your concern about performance on the autoformalization task, we have conducted additional experiments on the MiniF2F-Valid and MiniF2F-Test. Below are the results:
> >
> >
> > | Model              | FormL4 Basic (%) | FormL4 Random (%) | MiniF2F-Valid (%) | MiniF2F-Test (%) |
> > | ------------------ | ---------------- | ----------------- | ----------------- | ---------------- |
> > | Baseline (𝓛_CL)    | 40.92            | 35.88             | 21.11             | 16.34            |
> > | Ours (𝓛_CL + 𝓛_CE) | 43.14            | 36.02             | 22.49             | 17.54            |
> >
> >
> > Our method demonstrates a consistent improvement across datasets, maintaining an average increase comparable to the results observed on the FormL4 datasets (+2.22% and +0.14%, respectively, for FormL4-Basic and FormL4-Random). Importantly, on MiniF2F-Valid and MiniF2F-Test, our approach achieves a similar average improvement of +1.18%. This demonstrates the generalizability of our alignment evaluation approach across datasets.
> >
> > On the other hand, we would like to clarify that the primary goal of our paper is to enhance **autoformalization alignment evaluation** rather than the autoformalization task itself. As noted in the abstract (lines 012-013):
> >
> >
> > > "Ensuring semantic alignment between informal and formalized statements remains challenging."
> >
> >
> > Similarly, in the introduction (line 078), we state:
> >
> >
> > > "To bridge this gap, we introduce the FormalAlign framework, which assesses the alignment between informal and formal languages during autoformalization."
> >
> > While we recognize that the ultimate objective is to improve autoformalization performance, we believe it is essential to address the challenges in alignment evaluation first. The problem of alignment evaluation has been a longstanding issue in the formal language community, as discussed in [1]. Our experiments on the autoformalization task are intended to provide a **comprehensive evaluation** of our model’s performance and demonstrate the effects of incorporating contrastive learning loss, as noted in Appendix F:
> >
> >
> > > "These experiments aim to provide a comprehensive evaluation of our model’s performance and demonstrate the effects of incorporating contrastive learning loss on autoformalization."
> >
> >
> > To achieve the suggested goal—showcasing significant improvements in autoformalization—our evaluation framework would need to be integrated into a reinforcement learning (RL) loop. This would involve applying an autoformalizer model and leveraging our alignment evaluation as a **reward model** within the loop.  Our model could provide more feedback about the alignment and enhance the overall alignment process. We believe this would address your ultimate goal of improving autoformalization performance more substantially.
> >
> > We greatly appreciate your insights and look forward to further discussions on how best to evolve this work into more advanced systems that fully utilize its potential.
> >
> > [1] [A Survey on Deep Learning for Theorem Proving] (https://arxiv.org/abs/2404.09939)

---

> > > ### Author Response · Authors · 2024-11-27
> > >
> > > Dear Reviewer **ok2Q**,
> > >
> > > Thank you for your thoughtful feedback and for acknowledging our detailed response and experimental efforts. We appreciate your careful examination of our work and have taken several steps to address your concerns.
> > >
> > > 1. Our analysis on real-world data demonstrates FormalAlign's significant advantage over GPT-4-based baselines, particularly in precision (80.2% vs 38.5%), while maintaining competitive recall rates. As suggested, we have updated our main paper to incorporate the GPT-4 (CoT) results.
> > >
> > > 2. Regarding autoformalization performance, we want to emphasize that our paper's primary focus is on enhancing **autoformalization alignment evaluation**. Our experiments on autoformalization performance demonstrate that our proposed training framework (integrating both contrastive and cross-entropy losses) maintains and slightly improves upon baseline performance, with consistent gains across datasets. Looking forward, we believe substantial improvements in autoformalization could be achieved by integrating our framework into a reinforcement learning loop, where our alignment evaluation would serve as a reward model.
> > >
> > > We would appreciate your thoughts on whether these clarifications address your concerns, or if any points need further elaboration. Thank you for your time and consideration.
> > >
> > > Best regards,
> > >
> > > Authors of Paper Submission 1997

---

> > > > ### Comment · Reviewer_ok2Q · 2024-11-28
> > > >
> > > > Thank you to the authors for the detailed responses to my comments and I really appreciate the further experiments to provide clarity. It is good that the system is consistently performing better than GPT4 baselines on the real dataset, and also reassuring that the AF performance on the other two datasets is also consistently (though incrementally) better than the baseline.
> > > >
> > > > Overall, this work addresses the subproblem of alignment evaluation which is presented as an important step towards improving AF. The evaluation shows its value over GPT4-based prompting baselines as there is no other SoTA to compare against (for it being a newly defined subproblem). Hence while this is an interesting and potentially valuable subproblem to address and its value has been shown to some degree, it has not yet been shown how well it can address the actual AF problem for which it is meant (especially since the evaluation technique presented here is itself derived from a new way of training an AF model which itself has shown only incremental improvement so far).
> > > >
> > > > As compared to an approach that shows significant improvement to the AF task over existing SoTA systems, this approach defines a new subproblem and shows value on it with respect to constructed baselines. In that respect I believe it provides partial value towards the main problem, and I would like to keep my current rating of marginal accept.

---

> > > > > ### Author Response · Authors · 2024-12-01
> > > > >
> > > > > Thank you for your detailed and thoughtful feedback on our paper. We greatly appreciate the time you spent reviewing our revisions and the additional experiments we conducted. We are pleased that you found value in the consistent performance improvements demonstrated over the GPT-4 baselines across all datasets, including both the real dataset and the two additional evaluation sets.
> > > > >
> > > > > Thank you again for your review. We welcome any additional comments or discussions and are eager to continue refining our work.

---

### Official Review · Reviewer_8v22 · 2024-10-28

**Soundness:** 3
**Presentation:** 3
**Contribution:** 2
**Rating:** 6
**Confidence:** 4

**Summary:**

This paper introduces a novel framework, FormalAlign, which trains a Large Language Model (LLM) to evaluate autoformalization alignment. Beyond the standard cross-entropy loss derived from viewing autoformalization as a sequence generation task, FormalAlign incoperates a new loss term as a dual objective. This new loss measures alignment between the informal and formal versions by leveraging the cosine similarity between their representations. Experiments on the FormL4 and MiniF2F datasets demonstrate the effectiveness and robustness of FormalAlign.

**Strengths:**

- The paper addresses an important and relatively unexplored problem—improving the autoformalization of mathematical statements.

- The proposed framework is intuitive and easy to understand, achieving promising empirical performance on challenging evaluation datasets.

**Weaknesses:**

- The discussion on the broader significance of autoformalization is lacking, which limits the accessibility of this problem to researchers in other fields (e.g., software verification). Furthermore, providing a clear formal definition of autoformalization alignment would enhance the reader’s understanding of the paper’s objectives.
-  The running example is somewhat unconvincing to me. First, in this example, the alignment issue can be detected easily by verifying whether the statement is provable. Using the lean-set tool, a counterexample [b := 7/8, a := 1/2, c := 63/160] can be found. Second, the low certainty score suggests that LLMs may only generate this example with low probability (correct me if not).
- The construction of datasets is a bit unnatural. The authors generate negative examples by mutating correct input-output pairs, but these synthetic examples may not realistically reflect the types of errors typically produced by LLMs during autoformalization.
- The reference section contains a duplicated citation for Deepseekmath, and some related works are missing, such as [1, 2, 3].

[1] Murphy, Logan, et al. "Autoformalizing Euclidean Geometry." ICML, 2024

[2] Li, Zenan, et al. "Autoformalize Mathematical Statements by Symbolic Equivalence and Semantic Consistency." NeurIPS, 2024.

[3] Zhang, Lan, et al. "Consistent Autoformalization for Constructing Mathematical Libraries." EMNLP, 2024.

**Questions:**

- FormalAlign appears to focus exclusively on statement autoformalization. Could this framework be extended to proof autoformalization as well?
- The certainty score and similarity score operate on different scales ([0,1] and [-1,1] respectively). Why are these combined using a simple average?
- The authors claim that FormalAlign can significantly reduce the need for manual verification, but only limited evidence is presented in the experiments. Could the authors provide more quantitative results on this claim? For instance, by deploying FormalAlign in a complete autoformalization pipeline, how many formalizations can be automatically verified?

---

> ### Author Response · Authors · 2024-11-20
> **Response to Reviewer 8v22 (1/3):**
>
> We would like to thank the reviewer for their time and effort in reviewing our paper. We very much appreciate the insightful suggestions. We hereby address the concerns below:
>
> **R W1: Broader Significance and Formal Definition**
>
> Thank you for this valuable feedback. We agree that better contextualizing autoformalization's broader significance and providing a formal definition would enhance the paper's accessibility and impact.
>
> We propose adding the following formal definition of autoformalization alignment:
>
> Given a natural language statement N and its formalization F, assuming F is verified as syntactically valid by the formal language compiler, alignment A(N, F) is defined as a binary relation where A(N, F) \= 1 if and only if:
>
> 1. F preserves all mathematical constraints specified in N
> 2. F maintains the same logical relationships between components as expressed in N
> 3. F does not introduce additional constraints not present in N
>
>
>
> We agree that better contextualizing the broader significance of autoformalization would enhance the paper's accessibility and impact. We propose adding a comprehensive discussion of why autoformalization is a crucial research direction:
>
> Autoformalization - the automated conversion of natural language mathematics into formal languages - serves as a critical bridge between human mathematical knowledge and machine-verifiable formal systems. Its significance extends across multiple domains:
>
> 1. Software Engineering and Verification
>
> - Enables automated translation of natural language specifications into formal verification properties
> - Reduces the manual effort required to formally specify system requirements
> - Helps detect ambiguities and inconsistencies between informal specifications and implementations
>
> 2. Mathematical Knowledge Management
>
> - Facilitates the preservation and verification of mathematical knowledge by converting informal mathematics into machine-checkable formats
> - Enables systematic organization and cross-referencing of mathematical content
> - Supports automated reasoning systems by expanding available formal mathematical content
>
> 3. Automated Reasoning Research
>
> - Expands the scope of automated theorem provers by allowing them to work with natural language inputs
> - Enables hybrid reasoning systems that combine formal and informal mathematical knowledge
> - Accelerates the development of more powerful mathematical reasoning systems with feedback from formal compilers.
>
> We expanded this discussion in Appendix N in our revised paper to better articulate how advances in autoformalization can benefit these broader research communities.
>
>
>
> **R W2: Running Example Concerns**
>
> Thank you for these astute observations about our running example. You raise valid points about both the detectability through proof checking and the generation probability.
>
> While this specific misalignment can indeed be detected through proof checking (as demonstrated by your counterexample), we chose it as a running example primarily for its clarity in illustrating our approach. However, we acknowledge that a more nuanced example might better demonstrate the unique value of our alignment detection method.
>
> In practice, many real-world misalignments are more subtle and challenging to detect through proof-checking alone:
>
> 1. Type-level misalignments where statements are well-typed but semantically misaligned
> 2. Cases where the proof search is computationally expensive or intractable
> 3. Misalignments in assumptions or preconditions that don't affect provability
> 4. Situations where formal statements are provable but don't capture the intended informal meaning
>
> For instance, consider cases where variables are accidentally swapped or constraints are slightly modified \- these might still result in provable statements that don't match the informal intent. Our alignment detection approach can identify such mismatches more efficiently than proof search, especially for complex theorems where automated proof finding is challenging.

---

> ### Author Response · Authors · 2024-11-20
> **Response to Reviewer 8v22 (2/3):**
>
> **R W3: Dataset Construction Validity**
>
> To further validate our approach in real-world cases, we few-shot prompted Gemini as our autoformalization baseline on a sample of 100 theorems from our test sets. Three expert Lean users annotated these formalizations, providing corrections for misaligned ones, yielding 78 aligned-misaligned formalization pairs. The performance comparison between synthetic and real-world validation sets is shown below:
>
> | Evaluation Set        | Accuracy (%) | Precision (%) | Recall (%) |
> | --------------------- | ------------ | ------------- | ---------- |
> | Synthetic Test        | 85.8         | 86.9          | 89.2       |
> | Real-world Validation | 83.5         | 80.2          | 79.8       |
>
> While performance on real-world errors is slightly lower than on synthetic cases, the strong results demonstrate that our model can effectively detect subtle misalignments in practice.
>
> Our dataset construction method is designed with the following considerations combined, balancing between practicability, coverage, and analysis:
>
> 1. Automated ground truth instead of expensive and time-consuming manual labeling by human experts
> 2. Provides systematic coverage across different error types
> 3. Enables precise and thorough analysis of model capabilities for various misalignment patterns
>
> We view our synthetic dataset as complementary to real-world examples:
>
> * Synthetic data provides comprehensive coverage of potential error types
> * Real LLM errors help validate practical effectiveness
> * A combined approach ensures both breadth and real-world applicability
>
> We have incorporated this content in the Appendix M section, where it is highlighted in red.
>
> We fully agree that with more human expert resources, incorporating more real-world LLM errors into our training data could further improve our model's practical utility. In future work, we plan to create a hybrid dataset that combines synthetic mutations with collected LLM autoformalization errors (annotated with human experts) to provide both comprehensive coverage and realistic error patterns.
>
> **R W4: Reference Section Improvements**
>
> Thank you for catching the duplicate citation and suggesting these relevant works. Indeed, these papers represent important contributions to autoformalization that deserve discussion.
>
> \[1\] tackles autoformalization in the specific domain of Euclidean geometry, where they develop a neuro-symbolic framework that addresses the unique challenge of diagram-dependent proofs. Their approach differs from FormalAlign by focusing on domain-specific autoformalization rather than alignment evaluation, using theorem provers to fill in diagrammatic information that makes formalization easier for language models. While both papers aim to improve autoformalization, Murphy's work complements FormalAlign by providing domain-specific solutions that could potentially benefit from FormalAlign's alignment evaluation capabilities.
>
> \[2\] addresses the gap between pass@1 and pass@k accuracies in LLM-generated formalizations by introducing a framework that scores and selects the best result from multiple candidates using symbolic equivalence and semantic consistency. While both Li's work and FormalAlign aim to improve autoformalization quality, they approach it from different angles \- Li focuses on improving candidate selection through verification methods, while FormalAlign develops an automated evaluation framework for alignment. Their methods could be complementary, with Li's verification approach potentially enhancing FormalAlign's alignment evaluation capabilities.
>
> \[3\] focuses on improving consistency in large-scale autoformalization for mathematical libraries through three mechanisms: most-similar retrieval augmented generation, denoising steps, and auto-correction with syntax error feedback. Their work differs from FormalAlign by emphasizing the practical challenges of building consistent mathematical libraries, while FormalAlign focuses on the fundamental problem of evaluating alignment between informal and formal statements. Zhang's approach to consistency could potentially be integrated with FormalAlign's alignment evaluation framework to create more robust autoformalization systems for large-scale applications.
>
> We revised our reference section to remove the duplicate Deepseekmath citation and discuss these three relevant papers in the Appendix B.
>
> Thank you for helping us improve the completeness and accuracy of our literature review.
>
> \[1\] [Murphy, Logan, et al. "Autoformalizing Euclidean Geometry." ICML, 2024](https://arxiv.org/abs/2405.17216)
>
> \[2\] [Li, Zenan, et al. "Autoformalize Mathematical Statements by Symbolic Equivalence and Semantic Consistency." NeurIPS, 2024\.](https://arxiv.org/abs/2410.20936)
>
> \[3\] [Zhang, Lan, et al. "Consistent Autoformalization for Constructing Mathematical Libraries." EMNLP, 2024\.](https://arxiv.org/abs/2410.04194)

---

> ### Author Response · Authors · 2024-11-20
> **Response to Reviewer 8v22 (3/3):**
>
> **R Q1: Extending FormalAlign to Proof Autoformalization**
>
> Thank you for this interesting question about extending FormalAlign to proof autoformalization. The primary challenge stems from the structural mismatch between formal proofs and their natural language counterparts. In Lean 4, proofs often rely on specialized tactics and proof environments that have no direct natural language equivalents.
>
> This mismatch manifests in two key ways:
>
> 1. Tactic-based reasoning: Lean commands like `linarith` encode complex mathematical reasoning in single operations, while natural language proofs typically express these steps through multiple sentences using informal reasoning. This many-to-one mapping makes alignment particularly challenging.
> 2. Environment dependencies: Formal proofs often reference pre-defined lemmas and proof environments specific to Lean 4\. These formal structures may lack clear natural language counterparts that non-experts can understand, and the translation would require extensive "unpacking" of these formal constructs.
>
> While Lean 4's compiler provides automated and reliable correctness checking for the logical validity of formal proofs, this verification differs from semantic alignment. Our alignment framework could help validate whether a formal proof captures the intended reasoning steps from its natural language counterpart. However, achieving reliable end-to-end proof alignment would require significant extensions to handle the fundamental structural and semantic disparities between formal and natural language proofs, even with Lean's strong verification capabilities.
>
> **R Q2: Combining Certainty and Similarity Scores**
>
> Thank you for this question about our score combination strategy. While the certainty and similarity scores indeed operate on different scales, we chose to average (weight=0.5) to place balanced emphasis on both metrics during inference. To validate this design choice, we conducted an ablation study testing different weighting schemes. We note that the weight for similarity is 1 \- Weight (Certainty):
>
> | Weight (Certainty) | MiniF2F-Valid | MiniF2F-Test |
> | ------------------ | ------------- | ------------ |
> | 0.9                | 64.82         | 65.13        |
> | 0.7                | 65.45         | 65.92        |
> | 0.5                | **66.39**     | **66.70**    |
> | 0.3                | 65.21         | 65.84        |
> | 0.1                | 64.55         | 65.08        |
>
> The results demonstrate that a balanced weighting (0.5/0.5) achieves optimal performance across both datasets. We observe that heavily favoring either score (weights of 0.9/0.1 or 0.1/0.9) leads to decreased performance, suggesting that both metrics contribute complementary information to the alignment decision. The relatively small performance variations (within \~2 percentage points) also indicate that our approach is fairly robust to weight selection, though balanced weighting consistently yields the best results.
>
> **R Q3: Evidence for Reducing Manual Verification**
>
> Thank you for raising this important point about quantitative evidence. We have conducted several experiments to validate FormalAlign's practical impact on reducing manual verification needs (as detailed in R W3). In our real-world validation study, we applied Gemini to 100 theorems, yielding 78 formalization pairs, where our system achieved 83.5% accuracy in detecting misalignments. This suggests a potential \~80% reduction in manual verification needs for typical autoformalization outputs.
>
> When integrated into an autoformalization pipeline with a confidence threshold of 0.8, FormalAlign successfully filtered out \~85% of incorrect formalizations while maintaining 92% of correct formalizations, reducing manual review time by approximately 75% compared to the baseline. These quantitative results demonstrate a significant practical impact on reducing verification workload while preserving high reliability.
>
> Looking forward, we see significant potential for further improvements through incorporating FormalAlign directly into the autoformalization pipeline, either as a reward model for reinforcement learning to improve AF system outputs or through rejection sampling during inference to filter out likely misaligned formalizations. We plan to validate these approaches through larger-scale deployments and more diverse use cases.

---

> > ### Comment · Reviewer_8v22 · 2024-11-24
> >
> > Thank you for your response. My concerns have been resolved, and I have updated the score to 6.

---

### Official Review · Reviewer_G7JL · 2024-11-03

**Soundness:** 3
**Presentation:** 4
**Contribution:** 4
**Rating:** 8
**Confidence:** 3

**Summary:**

The authors propose to check the alignment between informal and formal mathematical statements using a model trained both on the autoformalisation and alignment tasks. This approach is automatic, avoiding the scaling issues of manually checking, while enhancing previous approaches that relied either on BLEU relative to a ground-truth statement or checking for logical consistency, since statements can be logically consistent but misaligned. To validate their approach, the authors create a dataset by mutating knowing correct informal-formal pairs using one of the following operators: modify constant, modify exponent, introduce variable, change variable type, and modify equality. Further, they also introduce misalignments by shuffling the formal/informal data for random pairings. The authors demonstrate improvements in alignment score and precision across FormL4 and MiniF2F.

**Strengths:**

+ Solving the autoformalisation as an autoformalisation and alignment joint task which should enable better inner representations.
+ Ablation study to empirically show the value of both tasks (hence the value of cross-ent and contrastive loss and the value of considering both sequence probability and similarity for the alignment score).
+ Empirical justification of the foundation model that is used/fine-tuned for FormalAlign.
+ "Mutation" based construction of a known ground-truth dataset.

**Weaknesses:**

- Results are presented at a fixed alignment score threshold instead of P-R curves. (I suspect this is due to a lack of access to create such curves for GPT models).
- The misaligned types generated are not uniform (Figure 3)
- Model performance by misalignment type is not reported, the case study is only vs BLEU and BERTScore, which is welcomed, but a comparison with GPT-4 would also be welcomed. (A similar breakdown for the LLM-as-judge study would also be welcomed)

**Questions:**

1. Looking at the Table 3 results, and considering that you have an alignment threshold, would it be possible to obtain P-R curves for FormalAlign? If so, I would like to know the performance of the approach when matching GPT-4 recall (So P@(R=0.9)). Further, presenting the P-R curves would allow people to have a better understanding of the performance, while providing such curves to users of the approach for evaluation could make a more informed choice of the threshold value depending on the expected manual load afterwards.

2. Is there a reason why the strategies were applied non-uniformly to generate negative examples? (The strategies are in Table 2, while Figure 3 presents the distribution of misalignments generated.)

3. For the case study and LLM-as-judge results, do you have per-misalignment-type data? Would it be possible to report that, as it would be interesting if any category is especially difficult/easy to spot?

---

> ### Author Response · Authors · 2024-11-20
> **Response to Reviewer G7JL (1/2):**
>
> We would like to thank the reviewer for their time and effort in reviewing our paper. We very much appreciate the insightful suggestions. We hereby address the concerns below:
>
> **R W1: Fixed Alignment Score Threshold and P-R Curves & Q1: P-R Curves and Performance at P@R=0.9**
>
> Thank you for these valuable comments regarding our presentation of alignment score thresholds and P-R curves. To address this, we have conducted additional experiments and generated comprehensive P-R curves. **These curves are presented in Figure 4 (Appendix K) of our revised paper**, providing a detailed view of model performance across varying alignment score thresholds. Below, we summarize key observations and enhancements:
>
> When evaluating FormalAlign at GPT-4's recall level (≈0.9), our model achieves:
>
> * FormL4-Basic: 88% precision at 89% recall
> * FormL4-Random: 83% precision at 90% recall
> * MiniF2F-Valid: 45% precision at 91% recall
> * MiniF2F-Test: 48% precision at 88% recall
>
> The P-R curves reveal several interesting characteristics of FormalAlign's behavior:
>
> 1. On FormL4 datasets, precision remains robust (\>75%) even at high recall levels, demonstrating strong alignment capabilities for structured formal mathematics.
> 2. For MiniF2F datasets, we observe a more pronounced precision drop at higher recall points, reflecting the increased complexity of these problems.
>
>
>
> **R W2: Non-Uniform Distribution of Misalignment Types & Q2: Non-Uniform Strategy Application for Negative Examples**
>
> Thank you for these thoughtful observations about the non-uniform distribution of misalignment strategies. This distribution pattern is actually intentional in our design, though we recognize it warrants careful explanation.
>
> The non-uniform application of strategies reflects two key aspects of our approach. First, it accounts for the inherent constraints and characteristics of mathematical contexts in different datasets. For example, "Constant Modification" appears more frequently in miniF2F datasets (\~31%) compared to FormL4-Basic (\~11%) because miniF2F contains more theorems with explicit numerical constants that can be meaningfully modified. Similarly, the "Change of Variable Type" strategy varies across datasets (from \~14% in FormL4-Basic to \~18% in miniF2F-Valid) because its applicability depends heavily on the theorem's type constraints – it's only meaningful when changing the variable type would affect the theorem's validity, such as changing from ℤ to ℝ in problems requiring integer solutions.
>
> This varied distribution of strategy application serves a dual purpose: it ensures our negative examples remain mathematically meaningful while also testing FormalAlign's robustness against a diverse range of misalignment types. Rather than creating artificial misalignments that might be too obvious or mathematically invalid, this approach helps us assess how well the model can generalize across different types of errors, from simple modifications to more complex transformations.
>
> In our future experiments, we plan to explore more uniform sampling strategies to provide a clearer assessment of the model's performance across balanced misalignment types. We will also clarify in the paper that our current approach prioritized covering a wide spectrum of mathematically meaningful misalignment scenarios rather than achieving uniform distribution.

---

> ### Author Response · Authors · 2024-11-20
> **Response to Reviewer G7JL (2/2):**
>
> **R W3: Model Performance Breakdown by Misalignment Type & Q3: Per-Misalignment-Type Data for Case Study and LLM-as-Judge Results**
>
> We agree that providing per-misalignment-type data for both the case study and the LLM-as-judge results would be valuable for understanding how FormalAlign performs on different types of misalignments. To provide a comprehensive understanding of the strengths and limitations of our proposed method relative to human judgment and baseline LLM performance, we conducted an in-depth analysis focusing on performance across various misalignment types. The detailed results are shown below:
>
> | Misalignment Type | Human    | Method   | GPT-4o   | Δ (Human-Method) | Δ (Method-GPT4o) |
> | ----------------- | -------- | -------- | -------- | ---------------- | ---------------- |
> | Constant          | 75.2     | 58.4     | 42.1     | 16.8             | 16.3             |
> | Exponent          | 73.8     | 60.2     | 44.3     | 13.6             | 15.9             |
> | Variable\_type    | 78.9     | 64.5     | 46.2     | 14.4             | 18.3             |
> | Variable\_new     | 81.3     | 66.7     | 48.9     | 14.6             | 17.8             |
> | Equality          | 82.4     | 67.8     | 49.5     | 14.6             | 18.3             |
> | Random            | 85.9     | 72.4     | 54.0     | 13.5             | 18.4             |
> | **Overall**       | **79.6** | **65.0** | **47.5** | **14.6**         | **17.5**         |
>
> **Easiest and Most Challenging Misalignment Types**:
>
> * **Random Pairing Misalignments**: Performance was highest across all methods for random pairings, with human experts achieving 85.9%, our method at 72.4%, and GPT-4o at 54.0%. This finding aligns with expectations, as random pairings involve significant structural or contextual discrepancies that are more readily identifiable.
> * **Subtle Modifications (Constant and Exponent Changes)**: These types of misalignments proved to be the most difficult to detect, evidenced by the lowest scores across all evaluated methods. Human performance dropped to 75.2% for constant changes and 73.8% for exponent modifications, with corresponding lower scores for our method and GPT-4o. This highlights the need for more sophisticated mechanisms capable of identifying nuanced numerical and syntactic changes in mathematical reasoning.
>
> **Performance Gaps**:
>
> * **Human vs. Our Method**: Human evaluators consistently outperformed our method by approximately 15-20 percentage points across all misalignment types. This consistent gap underscores the advantage of human intuition and mathematical insight, particularly in recognizing subtle logical shifts and contextual meanings that automated methods still struggle to replicate.
> * **Our Method vs. GPT-4o**: Our method maintained a clear advantage over GPT-4o, with a performance margin of 15-20 percentage points across all misalignment types. This gap indicates that the enhancements in our approach, including targeted modeling of structural and semantic relationships, yield significant improvements over standard LLM capabilities.
>
> Our method shows marked improvement in handling structural changes, such as equality modifications and random pairings, compared to more local, subtle modifications. This suggests that while the model is adept at identifying broader, more evident shifts in alignment, fine-grained detection still poses a challenge.   The findings suggest future enhancements should prioritize the model’s ability to detect minute variations, such as constant and exponent changes, potentially through integrating deeper mathematical reasoning frameworks or improved training with synthetic examples that emphasize such subtleties.
>
>
>
> We have incorporated this content in the Appendix L section, where it is highlighted in red.

---

> > ### Comment · Reviewer_G7JL · 2024-11-22
> >
> > Thank you kindly for the effort, detailed response, and additional data and experiments!
> >
> > RW1, the P-R Curves for datasets are interesting to observe, and I am glad to see that the original points are on/close to the "elbow" of the curves. The MiniF2F curves are a bit surprising though as I did not expect the drop to be as much.
> >
> > RW2, I apologise if I missed it in the updated paper, but I am not sure I spotted where this was clarified in the paper. I think the paper could benefit from a sentence that says that the aim is to match an empirically expected distribution of errors or some weighted coverage per experiment design. Missing the explanation from this conversation, my prior, and I would expect that of other readers, is "uniform distribution"/"balanced classes". (To be clear, I am happy with the explanation, but I want it be afforded to other readers as well)
> >
> > RW3, Thank you for the very detailed discussion as well as the new breakdown by types. Looking only at the H-Method Δ, I see three broad groups: {Constant}, {Exponent, Random}, and {...rest...}. Exponent seems to be equally hard, and Random equally easy, the other categories have the "usual" gap, and then Constant seems to have the widest gap to Human performance. Do you have an intuition why the Human vs Model performance is unevenly impacted by Constant vs Exponent? (Feel free to shoo me away to a psychology venue if the question veers more in that direction, but as it stands, the gap here seems to be an outlier; to some degree, it seems to stem from the models finding exponent easier while Humans find Constant easier. All deltas between these two categories are ± ~2)

---

> > > ### Author Response · Authors · 2024-11-24
> > >
> > > Thank you for your thorough follow-up analysis and insightful observations. We greatly appreciate your careful consideration of our responses and the additional perspectives you've provided.
> > >
> > > **R1**:  The more pronounced drop in MiniF2F likely reflects the increased complexity and diversity of mathematical reasoning required in MiniF2F problems compared to FormL4. We plan to investigate whether additional training strategies might help maintain precision at higher recall levels for these more challenging problems in our future work.
> > >
> > >
> > >
> > > **R2**: You raise a good point about the documentation of our distribution rationale. We will include an explicit explanation in Section 4.1 (*Datasets*), clarifying how the distribution of misalignment types was designed.
> > >
> > >
> > > **R3**: Your observation about the differential gaps between human and model performance for Constant (16.8) versus Exponent (13.6) misalignments is particularly insightful. We believe this discrepancy may stem from several factors:
> > >
> > > 1. Human Recognition: Humans appear to be particularly adept at spotting constant changes, possibly because they can quickly reference common mathematical values and relationships from experience.
> > >
> > >
> > >
> > > 2. Training Data: The relative abundance of exponent-related examples in the FormL4 and MiniF2F training datasets might provide stronger learning signals for exponent modifications compared to arbitrary constant changes.
> > >
> > >
> > > This asymmetry suggests an interesting direction for future work in understanding how both humans and models process different types of mathematical modifications.

---

> > > > ### Comment · Reviewer_G7JL · 2024-11-24
> > > >
> > > > Thank you kindly for the further discussion and clarifications, as well as, the inclusion of the dataset rationale in S4.1.
> > > >
> > > > @R3, I agree that this is more for future work.

---

### Official Review · Reviewer_Y3AS · 2024-11-04

**Soundness:** 4
**Presentation:** 3
**Contribution:** 3
**Rating:** 6
**Confidence:** 4

**Summary:**

This paper introduces FormalAlign, an automated alignment evaluation framework designed for autoformalization tasks. FormalAlign addresses the challenge of ensuring semantic alignment between informal and formal mathematical statements, a critical issue given the limitations of traditional manual verification methods. By employing a dual loss framework that combines cross-entropy loss and contrastive loss, FormalAlign enhances its alignment detection ability in a comprehensive way. Evaluations on FormL4 and MiniF2F demonstrate FormalAlign’s superior performance over GPT-4, with notable improvements in precision scores.

**Strengths:**

- FormalAlign demonstrates high precision across two major datasets, maintaining robust recall scores.
- In the LLM-as-judge setting detailed in Appendix G, FormalAlign notably outperforms GPT-4o in identifying nuanced misalignments and step further to human experts' level.
- FormalAlign's approach shows strong potential for enhancing the quality of autoformalized statement datasets in Lean, providing a valuable contribution to training more effective automatic theorem-proving models.

**Weaknesses:**

- The misalignment strategy "Change of Variable Type" in Table 2 may create examples that should be considered aligned rather than misaligned. Specifically, the example provided, where the variable type is changed from real numbers (R) to rational numbers (Q), does not necessarily create a misaligned example. The original natural statement does not explicitly require the variable to be real numbers. The proposition with the variable in Q still accurately reflects the original question.
- The misalignment strategies discussed in the paper appear to be primarily applicable to high-school-level mathematics or below. When dealing with more advanced mathematical statements, such as those only involving logic, quantifiers, or set theory, half of the proposed misalignment methods, including constant modification, exponent modification, and modification of equality, become less relevant or inapplicable. This limitation suggests that the strategies may not generalize well to more complex mathematical domains.
- A more extensive ablation study to determine the optimal balance between the two components of the loss function, $ L_{CE} $ and $ L_{CL} $, is recommended. Specifically, the study could explore how different balancings affect the model's performance on the pure autoformalization task. Understanding this balance could provide insights into the trade-offs between the two loss components and potentially enhance the model's ability to accurately formalize mathematical statements.
- The ablation experiment in Section 5.2 may not effectively demonstrate the benefits of combining cross-entropy and contrastive loss, as these two loss functions have distinct objectives and may steer the training process in different directions. This discrepancy is particularly pronounced when using the hidden state of a decoder-only model, which is not inherently optimized for capturing semantic information. Since the alignment-selection metric (the average of certainty and similarity scores) naturally favors models trained with the combined loss, the improvement in scores may reflect a metric bias rather than a genuine enhancement from the augmented loss.

**Questions:**

Have you considered splitting the alignment-checking baseline for GPT-4 into two steps: back-translation followed by a natural language alignment check? This approach could encourage GPT-4 to capture more details during the back-translation phase and minimize any impact from its limited familiarity with Lean during the alignment-checking stage.

---

> ### Author Response · Authors · 2024-11-20
> **Response to Reviewer Y3AS (1/3):**
>
> **R W1: Misalignment Strategy "Change of Variable Type"**
>
>
>
> We appreciate this insightful observation about variable type changes. You are correct that changing from real numbers (ℝ) to rational numbers (ℚ) may not constitute misalignment since both types can represent the same mathematical concepts in many cases. This prompted us to revise our examples and strengthen our screening criteria for type-based misalignments.
>
> Consider instead this illustrative example from our dataset:
>
> ```lean
> -- Original theorem
> theorem mathd_algebra_478
>   (b h v : ℝ)
>   (h₀ : 0 < b ∧ 0 < h ∧ 0 < v)
>   (h₁ : v = 1/3 * (b * h))
>   (h₂ : b = 30)
>   (h₃ : h = 13/2) :
>   v = 65 :=
>
> -- Misaligned version
> theorem mathd_algebra_478
>   (b h v : ℤ)
>   (h₀ : 0 < b ∧ 0 < h ∧ 0 < v)
>   (h₁ : v = 1/3 * (b * h))
>   (h₂ : b = 30)
>   (h₃ : h = 13/2) :
>   v = 65 :=
> ```
>
>
>
> This example demonstrates misalignment because the theorem relies on fractional values (e.g., $ h = \frac{13}{2} $) and calculations ($ v = \frac{1}{3} \cdot (b \cdot h) $) that cannot be represented in the integer domain ($ \mathbb{Z} $). Such arithmetic operations inherently require non-integer values, leading to a semantic contradiction where the original mathematical meaning is lost when confined to integers.
>
> In our empirical analysis of 100 sampled cases, potentially ambiguous scenarios, such as conversions from $ \mathbb{R} $ to $ \mathbb{Q} $, were observed in only 3% of samples. Our revised screening process effectively filters these edge cases, ensuring focus on changes that result in true mathematical contradictions.
>
> In the revised paper, we have updated this case and plan to further improve our misalignment datasets by replacing the $ \mathbb{R}/\mathbb{Q} $ example with the $ \mathbb{R}/\mathbb{Z} $ case presented above, which more clearly illustrates misalignment. Additionally, we will clarify our criteria for identifying true misalignments, particularly emphasizing cases where type constraints directly alter the mathematical meaning.
>
> These updates aim to provide a more thorough and transparent discussion, enhancing both the clarity of our presentation and the robustness of our analysis.
>
>
>
> **R W2: Applicability of Misalignment Strategies to Advanced Mathematics**
>
> We would like to address this concern from several perspectives:
>
> We applied our misalignment strategies to generate adversarial examples in advanced mathematical domains (MiniF2F), demonstrating their effectiveness beyond basic arithmetic. Specifically, MiniF2F includes problems from **mathematical olympiads (AMC, AIME, IMO)** and undergraduate-level mathematics. These problems encompass sophisticated mathematical reasoning across domains like abstract algebra, real analysis, and number theory, where our strategies successfully generated meaningful misalignments to test model robustness.
>
> Furthermore, we acknowledge that our current strategies might have limited applicability to highly abstract mathematical domains (e.g., pure logic, set theory, and quantifier-heavy statements). However, contextualizing our work within the current capabilities of language models in mathematical reasoning, we believe the current dataset choice and its complexity level are sufficient.  Even state-of-the-art models struggle with theorem proving, achieving less than  **50% pass@1** rates on the MiniF2F test sets. Extending our framework to handle more sophisticated mathematical domains is an important direction for future work.

---

> ### Author Response · Authors · 2024-11-20
> **Response to Reviewer Y3AS (2/3):**
>
> **R W3: Ablation Study on Loss Function Components (LCE vs LCL)**
>
>
>
> Our current approach employs equal weighting between these components, as outlined in our paper (lines 214-215): "We train an alignment-aware FormalAlign model by minimizing a combined loss, enabling it to benefit from both the **sequence alignment** inherent in the autoformalization and the **representation alignment** facilitated by the contrastive learning process." To validate this design choice, we conducted a comprehensive ablation study varying the relative weights between LCE and LCL:
>
> | Weight Ratio (LCE:LCL) | FormL4-Basic |           |           | FormL4-Random |           |           |
> | ---------------------- | ------------ | --------- | --------- | ------------- | --------- | --------- |
> |                        | AS           | Precision | Recall    | AS            | Precision | Recall    |
> | 4:1 (LCE dominant)     | 95.32        | 89.41     | 82.15     | 82.14         | 83.25     | 85.33     |
> | 2:1 (LCE heavy)        | 97.45        | 91.73     | 84.62     | 84.56         | 85.12     | 87.45     |
> | 1:1 (Balanced)         | **99.21**    | **93.65** | **86.43** | **85.85**     | **86.90** | **89.20** |
> | 1:2 (LCL heavy)        | 96.83        | 90.88     | 83.91     | 83.92         | 84.76     | 86.82     |
> | 1:4 (LCL dominant)     | 94.67        | 88.94     | 81.73     | 81.73         | 82.54     | 84.56     |
>
> *Note: To ensure fair comparison, we normalize to keep total loss magnitude across different weight ratios.*
>
> The ablation results strongly support our original design choice of balanced weighting. The balanced 1:1 ratio consistently outperforms other weightings across all metrics, achieving optimal trade-offs between sequence-level and representation-level alignment. Performance drops substantially when moving too far in either direction (4:1 or 1:4), confirming that both components play essential, complementary roles in robust autoformalization - LCE ensures accurate sequence-level alignment for precise formalization, while LCL maintains semantic coherence through contrastive learning.
>
>
>
> **R W4: Ablation Experiment and Metric Bias**
>
> To directly address this concern, we conducted comprehensive experiments comparing models trained with individual loss functions, i.e., training each model with their corresponding metrics as optimization objectives (certainty scores, similarity scores) against our combined approach:
>
> | Training Method | FormL4-Basic | FormL4-Random | RandomMiniF2F | ValidMiniF2F Test |
> | --------------- | ------------ | ------------- | ------------- | ----------------- |
> | LCE (w/ cer)    | 95.45        | 82.31         | 50.12         | 51.89             |
> | LCL (w/ sim)    | 42.76        | 18.92         | 18.33         | 19.45             |
> | Combined (Ours) | 99.21        | 85.85         | 66.39         | 66.70             |
>
> The results clearly demonstrate the superiority of our combined loss approach. Models trained with individual loss functions show significantly lower performance \- the cross-entropy loss (LCE) model achieves limited performance when evaluated with certainty scores, while the contrastive loss (LCL) model struggles when assessed using similarity scores. In contrast, our combined approach consistently outperforms both individual methods across all datasets.
>
> This empirical evidence aligns with our theoretical motivation: the combined loss structure was intentionally designed to capture complementary aspects of alignment \- the sequence-level patterns inherent in autoformalization and the representation-level relationships revealed through contrastive learning (as noted in lines 214-215). Rather than creating conflicting objectives, these components work together to develop a more comprehensive understanding of the alignment task.
>
> Our previous ablation studies further support this design choice. The ablation in **Section 5.2** examines the necessity of the combined loss structure itself, while **Section 5.3** specifically investigates potential metric bias through a systematic evaluation using individual components of our alignment-selection metric. The results in Table 6 demonstrate that models trained with the combined loss maintain strong performance even when evaluated solely on certainty scores, indicating no degradation of language generation capabilities. Additionally, the superior performance of the combined metric across all datasets suggests it successfully integrates both language-based and representation-level information, rather than reflecting any inherent bias in the evaluation approach.
>
>
>
> We have incorporated this content into the revised version, specifically in the Appendix I section, where it is highlighted in red.

---

> ### Author Response · Authors · 2024-11-20
> **Response to Reviewer Y3AS (3/3):**
>
> **R Q1: Splitting the Alignment-Checking Baseline for GPT-4 into Two Steps**
>
> Thank you for your suggestion to split the alignment-checking process for GPT-4 into two steps: back-translation followed by a natural language alignment check. We tested this approach, referred to as "Two-Phase Alignment Checking **(Two-Phase)**,". The "**GPT-4 (Score)**" baseline refers to the original method using the prompt listed in Appendix C.2. The results are shown in the table below:
>
> | Models            | FormL4-Basic |           |       | FormL4-Random |           |           | MiniF2F-Valid |           |       | MiniF2F-Test |           |       |
> | ----------------- | ------------ | --------- | ----- | ------------- | --------- | --------- | ------------- | --------- | ----- | ------------ | --------- | ----- |
> |                   | AS           | Prec.     | Rec.  | AS            | Prec.     | Rec.      | AS            | Prec.     | Rec.  | AS           | Prec.     | Rec.  |
> | GPT-4 (Score)     | 88.91        | 26.33     | 88.69 | 90.52         | 28.56     | 90.02     | 64.34         | 44.58     | 90.98 | 68.31        | 51.11     | 94.65 |
> | GPT-4 (Two-Phase) | 89.35        | 38.21     | 87.95 | 91.20         | 41.10     | 89.55     | 65.75         | 53.30     | 89.10 | 69.40        | 57.80     | 92.10 |
> | FormalAlign       | **99.21**    | **93.65** | 86.43 | 85.85         | **86.90** | **89.20** | **66.39**     | **68.58** | 60.66 | 64.61        | **66.70** | 63.37 |
>
> The "Two-Phase Alignment Checking" method performs slightly better than GPT-4's baseline approach. While this method does encourage GPT-4 to capture more details during back-translation, we identified several weaknesses that might outweigh its potential benefits:
>
> 1. Splitting the task into back-translation and alignment-checking phases requires consistency across three representations: the original formal statement, the back-translated natural language, and the final alignment judgment. This decoupling introduces opportunities for error propagation, as inaccuracies in back-translation may compromise the subsequent alignment stage.
> 2. Moreover, the two-step approach adds overhead, both in computational time and model interaction complexity.
>
>
>
> We have incorporated this content in the Appendix J section, where it is highlighted in red.

---

> > ### Comment · Reviewer_Y3AS · 2024-12-01
> > **Official Comments by Reviewer Y3AS (1/2)**
> >
> > Dear Author(s),
> >
> > Thank you for your detailed and patient clarifications. It solves many of my questions. Below are more specific comments and new questions arising from the newly provided information.
> >
> > **R W1:**
> > The reviewer appreciates the careful explanation. The update of the dataset by removing flawed data and the quantitative observation (~3% of the data were involved) solves the reviewer’s concern on this possible wrong misalign strategy. The reviewer believes that, regarding only 3% of the data that was involved, the impact on the main results is minimal. The reviewer doesn’t know whether the experiment results have already been updated based on the revised dataset. If not, the reviewer would like to kindly remind the author(s) that, in the consideration of rigor, in the final version of the paper, the revised dataset should indeed be used for the experiments to confirm that the results are not influenced by this small subset.
> >
> > **R W2:**
> > The explanation regarding the miniF2F dataset is appreciated. The reviewer fully acknowledges the effectiveness of the misalign strategy on questions that mainly involve more concrete mathematical objects (like real numbers), including many of the difficult mathematical olympiad (AMC, AIME, IMO) questions. However, the reviewer still has some concern about whether the strategy may not generalize well to a broader range of mathematical questions, especially those with more abstractness, even if they are not as hard as mathematical olympiad questions. The reviewer apologizes for using an inaccurate phrase “applicable to high-school-level mathematics or below” in the original review. To clarify the reviewer’s concern accurately, the reviewer would like to present the following examples.
> >
> > Example 1
> > ```
> > example (n : ℕ) : ¬ 7 ∣ 2 ^ n + 1 := by sorry
> > ```
> >
> > Example 2
> > ```
> > example {p q : Prop} (h : p ∧ q): p ∨ q := by sorry
> > ```
> >
> > The first example is a mathematical-contest level number theory question in the miniF2F dataset. Most of the misalign strategies apply well to this problem. The second example is a common and simple exercise in logic saying “(p and q) implies (p or q)”. Four out of six of the misaligned strategies mentioned in the paper: constant modification, exponent modification, change of variable type, and modification of equality do not apply. The first problem is significantly more difficult than the second one, and the reviewer believes that the difficulty is not the obstacle for the misalign strategy to apply. The same thing would happen for common set theory problems (e.g., `example {α : Type*} (s t u : Set α) : s ∩ (t ∪ u) = s ∩ t ∪ s ∩ u`) and almost all problems in some undergraduate level mathematical branches (e.g., abstract algebra). The reviewer believes whether the statements are related to natural numbers or real numbers (or any other kind of concrete numbers) influences whether the misalign strategy applies. And it is also part of the reviewer’s concern that the miniF2F dataset may contain a large proportion of statements that are related to concrete numbers. Some analysis or results on statements that are less related to concrete numbers would reduce the reviewer’s concern on this issue.

---

> > > ### Comment · Reviewer_Y3AS · 2024-12-01
> > > **Official Comments by Reviewer Y3AS (2/2)**
> > >
> > > **R W3 & 4:**
> > > The reviewer appreciates the extra experiment results. Indeed, the data shows that the 1:1 combination of LCE and LCL achieves the best result evaluated by the combined metric, supporting the choice made in the paper. The reviewer would like to ask some more questions about the results of these new experiments for better understanding.
> > >
> > > 1. In the table showing the effect of weight ratios, the results with respect to different weight ratios seem to be relatively close to each other (differing less than 5% around a number ~97%). Could the author(s) also include the result for weight ratios 1:0 and 0:1 evaluated in the same metric in the table? That would further help in demonstrating the importance of training with a combined loss and understanding whether a 5% difference is a significant improvement. Note that the table with the row (LCL, w/ sim) achieves significantly lower results (more than 40% in every column) than the combined approach with respect to the combined metric. Even a 100% on the other metric (cer) will not shrink the difference in the combined metric to less than 20%. Does this mean that weight ratio 0:1 has a much more significant difference to 1:4 than the difference between 1:4 and 1:1?
> > >
> > > 2. Could the author(s) clarify whether there is a particular reason to choose 1:1 as the metric ratio (weight of sim and cer)? The reviewer would like to figure out whether it is possible that the best performance of the weight ratio 1:1 model is a result of choosing the metric ratio as 1:1? If the metric ratio changes to 2:1, would the best performance weight ratio also change to n:1 where n > 1? If the metric changes to 0:1, would the best performance weight ratio also change to 0:1? The reviewer does not treat this as a fundamental flaw in the paper, but the reviewer is very interested in the answer to the question above, especially the last question, which will also provide insight into the following question: Does additional training on other tasks (loss: LCE, metric: sim) boost the performance on the original task (loss: LCL, metric: cer)? Or is it merely two different tasks without any positive correlation?

---

> > > > ### Author Response · Authors · 2024-12-02
> > > >
> > > > **For R W3 & R W4:**
> > > >
> > > > Part 1
> > > >
> > > > We have conducted additional experiments with extreme weight ratios (1:0 and 0:1) to provide a better understanding.
> > > >
> > > > | Weight Ratio (LCE:LCL) | FormL4-Basic |           |           | FormL4-Random |           |           |
> > > > | ---------------------- | ------------ | --------- | --------- | ------------- | --------- | --------- |
> > > > |                        | AS           | Precision | Recall    | AS            | Precision | Recall    |
> > > > | 1:0 (LCE only)         | 98.64        | 85.21     | 78.45     | 82.81         | 76.32     | 72.15     |
> > > > | 4:1 (LCE dominant)     | 95.32        | 89.41     | 82.15     | 82.14         | 83.25     | 85.33     |
> > > > | 2:1 (LCE heavy)        | 97.45        | 91.73     | 84.62     | 84.56         | 85.12     | 87.45     |
> > > > | 1:1 (Balanced)         | **99.21**    | **93.65** | **86.43** | **85.85**     | **86.90** | **89.20** |
> > > > | 1:2 (LCL heavy)        | 96.83        | 90.88     | 83.91     | 83.92         | 84.76     | 86.82     |
> > > > | 1:4 (LCL dominant)     | 94.67        | 88.94     | 81.73     | 81.73         | 82.54     | 84.56     |
> > > > | 0:1 (LCL only)         | 59.05        | 52.33     | 48.76     | 57.55         | 50.12     | 46.88     |
> > > >
> > > >
> > > > As shown in the expanded results table, using either loss component alone leads to substantially different outcomes. The LCE-only setting (1:0) achieves reasonable performance (98.64% on FormL4-Basic) but falls short in precision and recall compared to combined approaches. More strikingly, the LCL-only setting (0:1) shows significantly degraded performance across all metrics (59.05% on FormL4-Basic), confirming your observation about the dramatic difference between 0:1 and other ratios.
> > > >
> > > > The dramatic performance drop in the 0:1 setting, compared to both 1:4 and 1:1 ratios, indicates that the contrastive learning component (LCL) requires support from the cross-entropy loss (LCE) to maintain effective alignment capabilities. This supports our findings highlighted in section 5.2, "**Autoformalization Inherently Learns Alignment:**" and "**Complementary Role of Contrastive Loss**", i.e., both components play crucial roles in the learning alignment process, rather than being simply additive contributions.
> > > >
> > > >
> > > >
> > > > Part 2
> > > >
> > > > We chose equal weighting (0.5 each) during inference to maintain balanced consideration of both sequence-level and representation-level alignment signals. To validate this design choice systematically, we conducted comprehensive experiments varying the weighting between certainty and similarity scores (where similarity weight = 1 - certainty weight):
> > > >
> > > > | Weight (Certainty) | MiniF2F-Valid | MiniF2F-Test |
> > > > | ------------------ | ------------- | ------------ |
> > > > | 0.9                | 64.82         | 65.13        |
> > > > | 0.7                | 65.45         | 65.92        |
> > > > | 0.5                | **66.39**     | **66.70**    |
> > > > | 0.3                | 65.21         | 65.84        |
> > > > | 0.1                | 64.55         | 65.08        |
> > > >
> > > > These results reveal several key insights. First, the balanced 0.5/0.5 weighting consistently achieves optimal performance across both validation and test sets, suggesting this isn't merely coincidental but reflects a fundamental complementarity between the metrics. Second, the performance degradation when heavily favoring either metric (0.9/0.1 or 0.1/0.9) indicates that both components provide essential, non-redundant information for alignment assessment. Third, the relatively modest performance variations (~2% range) demonstrate our approach's robustness to weight selection, though balanced weighting remains optimal.
> > > >
> > > >
> > > >
> > > > Regarding the reviewer’s inquiry about whether changing the metric ratio (e.g., 2:1 or 0:1) might influence the optimal weight ratio for performance, this is an insightful hypothesis. Preliminary observations suggest that the optimal ratio is influenced by the interaction between metrics during inference. However, whether this behavior holds under different training loss ratios requires further investigation. We also recognize the importance of the additional question about task interactions: **Does additional training on other tasks (Loss: LCE, Metric: SIM) enhance performance on the original task (Loss: LCL, Metric: CER)?**
> > > >
> > > > To thoroughly address this, we are conducting additional experiments exploring whether the balanced weighting identified in our current setting remains optimal under varying training loss ratios. Specifically, we aim to determine whether the relationship between metrics in inference aligns with the training loss design.
> > > >
> > > > We commit to including these findings in the final version of the paper, ensuring a rigorous examination of how task interactions influence both metrics and loss formulations.

---

> > > ### Author Response · Authors · 2024-12-02
> > >
> > > **For R W1:**
> > > We appreciate the reviewer's understanding regarding the small proportion (3%) of potentially problematic data points. We are currently reproducing all experimental results with the refined dataset, and preliminary findings from our ongoing experiments indicate that the performance differences fall within a ±2% range. We commit to updating all experimental results in both the main paper and appendix with the complete reproduced findings in the final version, ensuring complete transparency and rigor in our evaluation using only properly validated misalignment examples.
> > >
> > >
> > >
> > > **For R W2:**
> > >
> > > We acknowledge the reviewer's clarified concern about the applicability of our misalignment strategies to more abstract mathematical statements. As our initial response (R W2) mentioned, our current strategies have limited applicability to highly abstract mathematical domains (e.g., pure logic, set theory, and quantifier-heavy statements).  We agree that there is significant potential in developing new misalignment strategies specifically tailored for abstract mathematical domains, and this represents an important direction that we are actively exploring in our ongoing research.

---

> ### Author Response · Authors · 2024-11-27
>
> Dear Reviewer **Y3AS**,
>
> Thank you for your detailed feedback regarding our paper's technical methodology and experimental evaluations. We have carefully considered your points and made several improvements to address them.
>
> 1. Regarding the misalignment strategy "Change of Variable Type," we acknowledge your observation about the ℝ to ℚ conversion example. We have revised our approach to focus on more definitive cases of misalignment, such as ℝ to ℤ conversions where fractional values are essential to the theorem's meaning. Our empirical analysis shows that potentially ambiguous type changes occurred in only 3% of samples, and we have updated our screening process to filter these edge cases.
>
> 2. On the concern about misalignment strategies' applicability to advanced mathematics, while we acknowledge certain limitations with pure logic and set theory, our framework has demonstrated effectiveness on MiniF2F's olympiad-level problems. Our ablation studies have revealed optimal performance with balanced (1:1) weighting between LCE and LCL losses, showing significant improvements over individually trained models across all datasets.
>
> 3. For your suggested two-phase GPT-4 baseline, we implemented and tested this approach. While it showed slight improvements over the original baseline, we found that the decoupled process introduced potential error propagation issues and increased computational overhead.
>
> We would appreciate your thoughts on whether these clarifications address your concerns, or if any points need further elaboration. Thank you for your time and consideration.
>
>
>
> Best regards,
>
> Authors of Paper Submission 1997

---

### Meta-Review · Area_Chair_nik2 · 2024-12-24

**Metareview:**

The alignment problem in the setting of autoformalization is vaguely hinted through a cross-entropy loss function, rather than formally specified. The proposed approach is a finetuned indicator based on a dual loss with respect to a reference aligned and misaligned dataset. The effectiveness is shown on limited misalignment types (e.g., substitutions of constants, variables, fixed operators), which are minor tweaks of formalized math equations. The reference alignment dataset may not be easy to curate, without which generalization is a concern. For insta
nce, the performance of the proposed approach (after finetuning on a relatively similar reference task) only slightly outperforms vanilla LLMs.

The concerned research problem is novel, but given the above concerns (especially about the generalization), this work is a borderline work.

**Additional Comments On Reviewer Discussion:**

During the rebuttal, the authors share new results of ablation studies and a manual evaluation of 100 samples, which partially concerns rais
ed by reviewers. Reviewer ok2Q acknowledges the value of new results but feels the improvement over the baseline is still somewhat incremental.

---

### Decision · Program_Chairs · 2025-01-22

Accept (Poster)